# Monitor-Guided Decoding of Code LMs with Static Analysis of Repository Context

**Lakshya A Agrawal**
Microsoft Research
Bangalore, India
t-lakagrawal@microsoft.com

**Aditya Kanade**
Microsoft Research
Bangalore, India
kanadeaditya@microsoft.com

**Navin Goyal**
Microsoft Research
Bangalore, India
navingo@microsoft.com

**Shuvendu K. Lahiri**
Microsoft Research
Redmond, United States
shuvendu.lahiri@microsoft.com

**Sriram K. Rajamani**
Microsoft Research
Bangalore, India
sriram@microsoft.com

## Abstract

Language models of code (LMs) work well when the surrounding code provides sufficient context. This is not true when it becomes necessary to use types, functionality or APIs defined elsewhere in the repository or a linked library, especially those not seen during training. LMs suffer from limited awareness of such global context and end up hallucinating.

Integrated development environments (IDEs) assist developers in understanding repository context using static analysis. We extend this assistance, enjoyed by developers, to LMs. We propose *monitor-guided decoding* (MGD) where a monitor uses static analysis to guide the decoding. We construct a repository-level dataset PRAGMATICCODE for method-completion in Java and evaluate MGD on it. On models of varying parameter scale, by monitoring for type-consistent object dereferences, MGD consistently improves compilation rates and agreement with ground truth. Further, LMs with fewer parameters, when augmented with MGD, can outperform larger LMs. With MGD, SantaCoder-1.1B achieves better compilation rate and next-identifier match than the much larger text-davinci-003 model.

We also conduct a generalizability study to evaluate the ability of MGD to generalize to multiple programming languages (Java, C# and Rust), coding scenarios (e.g., correct number of arguments to method calls), and to enforce richer semantic constraints (e.g., stateful API protocols). Our data and implementation are available at https://github.com/microsoft/monitors4codegen.

## 1 Introduction

Language models of code (LMs), such as in Chen et al. (2021); Nijkamp et al. (2023); Allal et al. (2023) and many others, are revolutionizing code generation. Many commercial offerings based on LMs, like GitHub Copilot, Amazon Code Whisperer and Replit, are now available. The LMs work well when the surrounding code in the vicinity of generation provides sufficient context. This

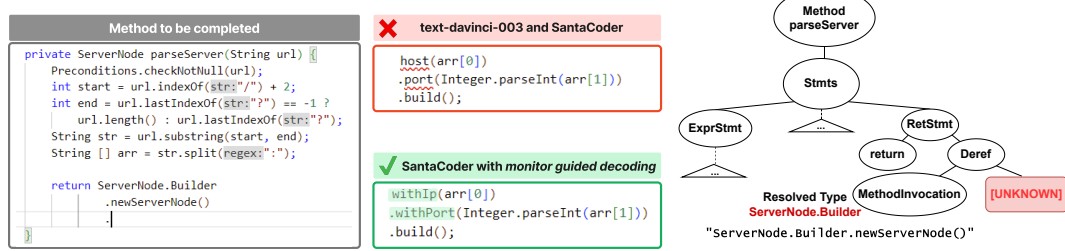

(a) Example where text-davinci-003 and SantaCoder generate wrong identifiers, but SantaCoder with MGD generates correct identifiers.

(b) Annotated partial AST for the code to the left.

Figure 1: Motivating example to illustrate monitor-guided decoding (MGD).

condition does not hold when it becomes necessary to use types, functionality or APIs defined in another module in the repository or an external library, especially those not seen during training. In the absence of awareness of such global context, the LMs end up hallucinating, e.g., using types defined in other files incorrectly. Further, devoid of the repository context, the LMs may lack awareness of other semantic information like the number of arguments required by a called method, or the ordering constraints on method calls (API protocols) to be followed. Often, such type and semantic information can come from artifacts generated at build-time, like Project Lombok (lom, 2009) and ProtoBuf (pro, 2008), and therefore may not even be present as code context in the repository.

As an example, consider the partial code (method to be completed) in Figure 1(a). To complete this code, an LM has to generate identifiers consistent with the type of the object returned by `ServerNode.Builder.newServerNode()`. The method `newServerNode` and its return type, class `ServerNode.Builder`, are defined in another file. If an LM does not have information about the `ServerNode.Builder` type, it ends up hallucinating.

We show a completion generated by the OpenAI text-davinci-003 (Ouyang et al., 2022) and Santa-Coder (Allal et al., 2023) models in the box decorated with ✖ in Figure 1(a). The completion uses identifiers `host` and `port`, which do not exist in the type `ServerNode.Builder`. The generated code therefore results in "symbol not found" compilation errors. The lack of awareness of other semantic information from the global context may result in other kinds of errors like compile-time errors (e.g.,"actual and formal argument lists differ in length" on using wrong number of arguments) or runtime errors (e.g., `IllegalStateException` on violation of API protocols).

Integrated development environments (IDEs) have been at the forefront of assisting developers. Our inspiration is the use of static analysis by IDEs to bring the global context at the fingertips of developers. Many analyses are integrated in IDEs (Fuhrer, 2013) to infer and enforce semantic constraints on the code under development, e.g., resolving def-use, symbol references, and type hierarchies. Recently, there has been a rise in the use of Language Server Protocol (LSP) (lsp), which is an open industry standard of communication between IDEs and programming language specific tools like static analyzers and compilers, called Language Servers. There are a large number of Language Servers available, targetting most programming languages (lan, 2023), and providing a variety of syntactic and semantic information. In this work, we focus on the type-directed code completion analysis available through LSP in a language-agnostic manner, to provide guidance to an LM.

We propose a notion of *monitors* as a stateful interface between LMs and static analysis. A monitor observes the code generated by an LM and queries static analysis at pre-defined trigger points. The suggestions returned by the static analysis are converted to masks which are used for reshaping the logits (or equivalently, token-generation probabilities) produced by the LM in the subsequent decoding steps. We call our method *monitor-guided decoding* (MGD). Unlike an LM, a static analysis operates on the entire repository and its dependencies. While the LM generates completions by conditioning on the local context, the static analysis ensures consistency with the rest of the code in the repository. Through MGD, we bring the two together without the need to retrain the LM, and making a minor and modular addition to the decoding stage of the LM.

Figure 1(a) also shows the code generated by the SantaCoder model with MGD in the box decorated with ✓. This code makes use of identifiers that are actually defined in the class

`ServerNode.Builder`. It compiles and matches the ground truth. In comparison, the *same* Santa-Coder model without MGD generates the erroneous code shown in the box decorated with ✖.

Some recent approaches use static analysis (Shrivastava et al., 2022; Ding et al., 2022; Pei et al., 2023) or retrieval (Zhang et al., 2023) to extract relevant code fragments from the global context. These approaches expand the prompt (Shrivastava et al., 2022; Pei et al., 2023; Zhang et al., 2023) or require architecture modifications (Ding et al., 2022) and additional training (Ding et al., 2022; Pei et al., 2023). In comparison, we provide token-level guidance to a frozen LM by invoking static analysis on demand. Our method is complementary to these approaches as they condition the generation by modifying the *input* to the LM, whereas we apply *output*-side constraints by reshaping the logits.

We make the following contributions in this paper:

- Monitor-Guided Decoding (MGD) as a stateful interface between LMs and static analysis. A monitor watches the LM generating code, queries static analysis in the background, and uses the information from the static analysis to effectively guide the decoding stage of LMs.
- PRAGMATICCODE, a publicly-released dataset of Java code repositories complete with their development environments and dependencies.
- Instantiation of MGD for generating code having type-consistent identifier dereferences.
- Large scale evaluation on PRAGMATICCODE showing: (1) MGD consistently improves the ability of an LM (across varying parameter scales) to generate type-consistent identifiers, compilation rates and agreement with ground truth. (2) Further, LMs with fewer parameters, in conjunction with MGD, can outperform larger LMs. For instance, SantaCoder-1.1B with MGD achieves better compilation rate and next-identifier match than the much larger text-davinci-003 model, when both have a budget of 1 generation each. With a budget of 4 generations, it also surpasses agreement with ground truth of text-davinci-003. (3) We also evaluate how MGD complements different prompt augmentation and decoding strategies.
- Microbenchmark to demonstrate generalizability of MGD to different (1) programming languages, (2) coding scenarios, and (3) use of other static analysis techniques for guiding with rich semantic properties. Notably, we demonstrate that small-LMs can be guided to adhere to rich static properties like satisfaction of stateful API protocols.
- We open source our implementation and provide an extensible Python library called `multilspy` with static analysis bindings for multiple languages over LSP, suitable for monitoring of LMs. It can be used for other AI4Code scenarios as well.

## 2 Monitor-Guided Decoding

**Background.** Static analysis of code (Nielson et al., 2015) is used widely in industry in various applications such as in detecting bugs and optimizing code. While analysis is usually performed on complete code, IDEs have long applied static analysis on incomplete code under development (Reps et al., 1983; Dagenais & Hendren, 2008), using algorithms for incremental parsing and semantic analysis (e.g., type inference) of partial code (Hedin, 1992; Wagner, 1997; Maddox III, 1997). These analyses have now become a standard part of language servers.

A key abstraction used in analysis of partial code is partial abstract syntax trees (ASTs) with special nodes to indicate incomplete parts of code. These ASTs are further decorated by semantic information through attribute grammars (Reps et al., 1983) that decorate each AST node with attributes that capture the static semantics not captured in the context-free grammar (such as consistency of types among expressions in an assignment). This can range from computing the type-hierarchy for object oriented languages, binding the variables in AST nodes to their declarations, resolving the types of expressions as well as computing the def-use relationships for resolved AST nodes (Fuhrer, 2013).

Figure 1(b) shows the partial AST for the incomplete code in Figure 1(a). All the statements upto the incomplete `return` statement are completely parsed and subtrees corresponding to them are constructed. The subtree for the `return` statement includes a node `[UNKNOWN]` indicating the incomplete part. As shown in Figure 1(b), an incremental semantic analysis resolves the type of the partial expression `ServerNode.Builder.newServerNode()` to `ServerNode.Builder`. Later, we show how to use this type information to construct a monitor, which can then be used to guide an LM to generate type-consistent identifier completions.

**Basic Concepts and Notation.** Consider an LM $L_\theta$ operating on a vocabulary $V$. Let $x_1, \ldots, x_n$ be the *partial code* that has been generated by the LM and $x_{n+1}$ be a candidate next token. Though a vanilla (auto-regressive) prompt would consist only of $x_1, \ldots, x_n$, today many approaches augment it with additional information. We use $p$ to indicate this *additional prompt*, e.g., the suffix information used in fill-in-the-middle prompting (Donahue et al., 2020; Fried et al., 2022; Bavarian et al., 2022).

A *property* $\varphi$ specifies the constraints that a piece of code needs to satisfy. A *static analysis* $A_\varphi$ checks whether the partial code satisfies $\varphi$. If yes, it returns suggestions to extend it so that the extended code continues to satisfy $\varphi$. The static analysis works on the *repository context* $C$ which not only includes code spread across multiple files in the repository, but also external dependencies and intermediate artifacts (e.g., code bindings) generated during the build process. Such repository context is often very large, diverse and complex. Directly including it as an input to the LM will result in bloating and pass the burden of distilling useful information from it to the LM.

A *monitor* $M_\varphi$ for a property $\varphi$ is a tuple $(A_\varphi, s_0, S, \mathtt{pre}, \mathtt{update}, \mathtt{maskgen})$. The monitor starts in the *wait state* $s_0$. If the partial code satisfies the *pre-condition* $\mathtt{pre}$ then the monitor is triggered and it invokes $A_\varphi$ on the partial code. The monitor maintains the suggestions returned by $A_\varphi$ in its state and uses them to guide sampling of the next token $x_{n+1}$. Let $S$ be the set of states and $s, s' \in S$ respectively be the current and next states of the monitor. With each sampled token $x_{n+1}$, the monitor *updates* its state using a function $\mathtt{update}(s, x_{n+1})$ to a new state $s'$, which tracks the residual suggestions after the token $x_{n+1}$ is output. When the suggestions are exhausted, it reverts to the wait state $s_0$. We explain the function $\mathtt{maskgen}$ below.

**Decoding Process.** Usually, the next token $x_{n+1}$ can be any token from the vocabulary $V$, sampled based on the logits $\ell$ determined by the LM. Unlike the usual decoding, in *monitor-guided decoding*, we supervise the code generation using a monitor $M_\varphi$ for a property $\varphi$. We denote the composition of $L_\theta$ and $M_\varphi$ by $L_\theta || M_\varphi$, meaning that both the LM and the monitor are running concurrently and sampling the tokens jointly. The decoding is conditioned on the partial code $x_1, \ldots, x_n$, the repository context $C$, the prompt $p$ and the current state $s$ of the monitor.

Eq. (1) states that whenever the monitor is in the wait state $s_0$, we sample $x_{n+1}$ as per the logits $\ell$ determined by the LM (Eq. (2)). Otherwise, the logits are combined with a mask $m$ using a function $\oplus$ such that if $m[x] = 0$ then $\ell[x]$ is reset to a large negative value $-K$ and is left unchanged otherwise. This mask is computed by the function $\mathtt{maskgen}$ in Eq. (3) guided by the current state $s$ of the monitor. Eq. (4) defines how the state of the monitor evolves. When the pre-condition $\mathtt{pre}(s; x_1, \ldots, x_n)$ evaluates to true, the next state $s'$ of the monitor is determined by the suggestions returned by the static analysis $A_\varphi$. Otherwise, it is determined by the $\mathtt{update}$ function.

$$(L_\theta || M_\varphi)(x_{n+1} | x_1, \ldots, x_n; C, p, s) = \begin{cases} \mathtt{softmax}(\ell)[x_{n+1}] & \text{if } s = s_0 \text{ is the wait state} \\ \mathtt{softmax}(\ell \oplus m)[x_{n+1}] & \text{otherwise} \end{cases} \quad (1)$$

$$\ell = L_\theta(\, \cdot \, | x_1, \ldots, x_n; p) \quad (2)$$

$$m = \mathtt{maskgen}(s, V) \quad (3)$$

$$s' = \begin{cases} A_\varphi(x_1, \ldots, x_n; C) & \text{if } s = s_0 \wedge \mathtt{pre}(s; x_1, \ldots, x_n) \\ \mathtt{update}(s, x_{n+1}) & \text{otherwise} \end{cases} \quad (4)$$

The specifics of the monitor state, and the $\mathtt{pre}$, $\mathtt{update}$ and $\mathtt{maskgen}$ functions depend on the static analysis $A_\varphi$ used by the monitor. Our formulation is general and even allows combining multiple static analyses by taking a product of the state-spaces of their respective monitors. In the following, we discuss a specific instantiation of this framework of monitor-guided decoding.

**Monitoring for Use of Type-Consistent Identifiers.** When an object $\mathtt{obj}$ of a type $T$ is dereferenced, the next token (or more generally, the sequence of subtokens) should refer to an identifier of a field or method defined by the type $T$. It can otherwise result in a "symbol not found" error. The type $T$ could be defined in another package, imported file or in a library. Unless $T$ comes from a popular library seen during training, the LM may not have knowledge about $T$.

Our monitor $M_\varphi$ is triggered when the partial code $x_1, \ldots, x_n$ ends with a partial object dereference expression "$\mathtt{obj.}$" where "$.$" is the dereference operation. This is the pre-condition $\mathtt{pre}$ we use. We employ a static analysis $A_\varphi$ which returns all the type-consistent identifiers that can be referenced through $\mathtt{obj}$. For this, $A_\varphi$ performs a global analysis over the partial code, imported files, libraries used, and class hierarchies to infer the type $T$ of $\mathtt{obj}$ and the identifiers accessible through $T$. The set of type-consistent identifiers returned by $A_\varphi$ forms the state of the monitor (see Eq. (4)).

Given a state $s$ and the vocabulary $V$ of the LM, `maskgen` identifies all the tokens in $V$ that are consistent with the suggestions in $s$. The identifiers returned by the static analysis are tokens as per the programming language, whereas the vocabulary $V$ may use its own space of subtokens (Schuster & Nakajima, 2012; Kudo & Richardson, 2018). The mask $m$ (Eq. (3)) is generated by string matching. For all tokens $t \in V$ that form prefixes of the identifiers in $s$, the mask value $m[t]$ is set to 1, indicating that they can be sampled. Let $E$ be the set of special symbols that indicate end of an identifier name, e.g., the symbol '(' in 'getName(' or ',' in 'x,'. Let $w$ be a string in $s$ and $\Sigma$ be the set of all possible characters. If a token $t \in V$ matches the regular expression $w \cdot E \cdot \Sigma^*$ then its mask value $m[t]$ is also set to 1. For all other tokens $t$ in $V$, the mask value $m[t]$ is set to 0.

Let $x_{n+1}$ be the next token sampled according to the second equation in Eq. (1). If $x_{n+1}$ contains a symbol from $E$, indicating that a complete identifier name has been generated in accordance with the set returned by $A_\varphi$, the monitor reverts to the wait state to wait for the next trigger. Otherwise, the token $x_{n+1}$ must be a prefix of a member of $s$. The `update` function removes the members in $s$ that are not prefixed by $x_{n+1}$, and those prefixed by $x_{n+1}$ are updated by pruning the prefix string $x_{n+1}$. The resulting set of character strings forms the next state $s'$ (see the second equation in Eq. (4)). If $A_\varphi$ returns an empty set to start with, we abandon the current run. Note that a single identifier may need to be generated using multiple tokens. Figure 4 (see Appendix A) shows the interaction between the LM and the monitor for the example in Figure 1, and specifically illustrates how the complete identifier names suggested by the static analysis are gradually pruned by the monitor to corresponding suffixes in each successive state as the prefixes get generated as tokens.

## 3   Experimental Setup

**Dataset Creation.**  In order to evaluate MGD, we need real-world repositories with their build environments and dependencies. Most published datasets are standalone, with the exception of CoderEval (Yu et al., 2023) and PyEnvs (Pei et al., 2023), both of which are not publicly available at the time of this writing. Hence we curated PRAGMATICCODE, a dataset of real-world open-source Java projects complete with their development environments and dependencies. We ensure that these repositories were released publicly only after the determined training dataset cutoff date (31 March 2022) of the models which we use to evaluate MGD.

From PRAGMATICCODE, we identify a set of method-level completion task instances, creating DOTPROMPTS as a method-level code completion benchmark. Each testcase in DOTPROMPTS consists of a prompt upto a dereference location (using the "." operator in Java) within a target method, and the task is to complete the remainder of the method. We ensure sufficient complexity in the identified target methods in DOTPROMPTS by including methods that satisfy a set of complexity filters (e.g., the method should have at least 7 lines of code) described in detail in appendix B. Overall, PRAGMATICCODE consists of **100** repositories, and DOTPROMPTS consists of **1420** methods and **10538** dereference prompts. Appendix B gives further details.

**Models.** We study the effect of performing MGD on code generation with the HuggingFace Transformers (Wolf et al., 2020) implementation of Salesforce CodeGen family of models (CodeGen-{350M, 2B, 6B}-Multi, abbreviated as **CG-{350M, 2B, 6B}** hereafter) (Nijkamp et al., 2023) and BigCode SantaCoder-1.1B (**SC** or SantaCoder hereafter) (Allal et al., 2023). We also evaluate OpenAI text-davinci-003 (**TD-3** hereafter) with and without MGD, available on Azure.

**Prompting Strategies.** We study the effect of different prompt augmentation techniques when combined with MGD: (1) **Standard**: Include the local file content up to the dereference point and truncate from left to fit the prompt budget. (2) **classExprTypes**: For a given target method belonging to a class `C`, identify the type of all expressions occurring in `C` (after masking out the target method to prevent leakage) and include the concatenated file contents for the type definitions of all the identified files, truncating from the left as necessary. We assign a budget of 20% tokens of total prompt budget to classExprTypes. (3) **RLPG**: Use the prompt augmentation technique proposed in Shrivastava et al. (2022). We use their released source code and model checkpoints to adapt RLPG to DOTPROMPTS.

**Decoding Strategies.** We experiment with two decoding strategies: (1) **Autoregressive:** for left-to-right decoding. (2) **Fill-in-the-middle:** Use fill-in-the middle (FIM) setting implemented in SantaCoder (Allal et al., 2023).

Table 1: Summary of results with a budget of 6 generations per model: The numbers in parentheses are relative improvements of the "-MGD" configuration over the respective base model.

| Config \ (Metric, score@6) | CR | NIM | ISM | PM |
|---|---|---|---|---|
| CG-350M | 52.43 | 76.94 | 31.86 | 26.86 |
| CG-350M-MGD | 65.37 (24.69%) | 83.80 (8.92%) | 34.31 (7.70%) | 28.69 (6.82%) |
| CG-2B | 57.01 | 81.11 | 36.38 | 30.95 |
| CG-2B-MGD | 70.91 (24.38%) | 87.32 (7.66%) | 39.03 (7.29%) | 33.06 (6.82%) |
| CG-6B | 58.64 | 81.55 | 37.17 | 31.69 |
| CG-6B-MGD | 72.28 (23.25%) | 87.35 (7.11%) | 39.55 (6.41%) | 33.56 (5.89%) |
| SC | 59.97 | 82.40 | 38.14 | 32.10 |
| SC-MGD | 73.03 (21.77%) | 88.42 (7.31%) | 40.69 (6.68%) | 34.25 (6.72%) |
| SC-classExprTypes | 64.57 | 84.91 | 39.67 | 33.55 |
| SC-classExprTypes-MGD | 75.01 (16.18%) | 89.37 (5.25%) | 41.56 (4.78%) | 34.98 (4.26%) |
| SC-RLPG | 66.39 | 85.42 | 42.35 | 36.21 |
| SC-RLPG-MGD | 78.14 (17.70%) | 89.89 (5.23%) | 44.47 (5.00%) | 37.97 (4.87%) |
| SC-FIM | 68.23 | 85.56 | 42.22 | 36.12 |
| SC-FIM-MGD | 80.19 (17.52%) | 89.89 (5.07%) | 44.50 (5.39%) | 37.91 (4.95%) |
| SC-FIM-classExprTypes | 70.97 | 86.99 | 42.67 | 36.36 |
| SC-FIM-classExprTypes-MGD | 80.33 (13.18%) | 90.42 (3.94%) | 44.18 (3.54%) | 37.75 (3.82%) |
| TD-3 | 62.66 | 86.18 | 44.97 | 38.77 |
| TD-3-MGD | 74.26 (18.52%) | 91.19 (5.81%) | 47.33 (5.24%) | 39.94 (3.03%) |

**Metrics.** We use the following metrics to measure the quality of generated code: (1) **Compilation Rate (CR):** We replace the ground truth method with the generated method in the context of the complete repository and invoke a clean build. We assign a score of 1 if the compilation succeeds, and 0 otherwise. (2) **Match with the ground truth:** We use three specific metrics to measure how closely the generation matches ground truth, namely (a): **Next Identifier Match (NIM):** If the first Java token generated by the LM matches with the ground truth, we assign a score of 1, 0 otherwise; (b) **Identifier Sequence Match (ISM):** Longest prefix match between the ordered set of identifier names in the ground truth and generated completion, normalized by the number of identifiers in the ground truth; and (c) **Prefix Match (PM):** Longest prefix match between the ordered set of Java tokens (as obtained from a Java Lexer) between the ground truth and generated completion, normalized by the number of tokens in the ground truth. Except NIM, all other metrics - namely CR, ISM and PM - evaluate the complete method-level generation by the model.

In our experiments, for all the evaluated model configurations, we use nucleus sampling (Holtzman et al., 2020) with a top-p value of 0.95 to generate $n = 6$ independent samples. For a budget of $k \in [1, n]$ samples, we compute the aggregate score $score@k$ (see Appendix D). On the discrete-valued metrics (CR and NIM), it is identical to $pass@k, n$ (Chen et al., 2021), estimating the expected number of times the list of $k$ candidates contains at least one successful compilation or match with ground-truth identifier. On the real-valued metrics (ISM and PM), it estimates the expectation of the maximum value of the corresponding metric given $k$ chances.

**Python Library for MGD.** We are releasing an extensible Python library, `multilspy`, for interfacing between LMs and language servers using LSP. It can be used with multiple programming languages, static analyses and LMs. It also supports an approximate mechanism for MGD of black-box LMs with limited logit-masking support. Please refer to Appendix C for more details.

## 4 Evaluation

Table 1 shows the summary of the results for all of our experiments and metrics. For the "-MGD" configurations, we also report the relative improvement over the *base model* in parentheses, where the base model is the same model configuration without MGD. Below, we present a detailed evaluation.

### 4.1 Effect of MGD on Models across Parameter Scale and Architectures

We present results for all the models on Standard prompts described in Section 3.

**Compilation Rate.** As shown in Figure 2a, all the base models with MGD, including the smallest model CodeGen-350M for $k \geq 4$, outperform the largest model text-davinci-003, by maximum relative margin of 16.55% achieved by SantaCoder. All the models with MGD outperform their respective base models on Compilation Rate, by a relative improvement ranging between 21.77%-24.69%. TD-3-MGD outperforms TD-3 by a relative margin of 18.52%.

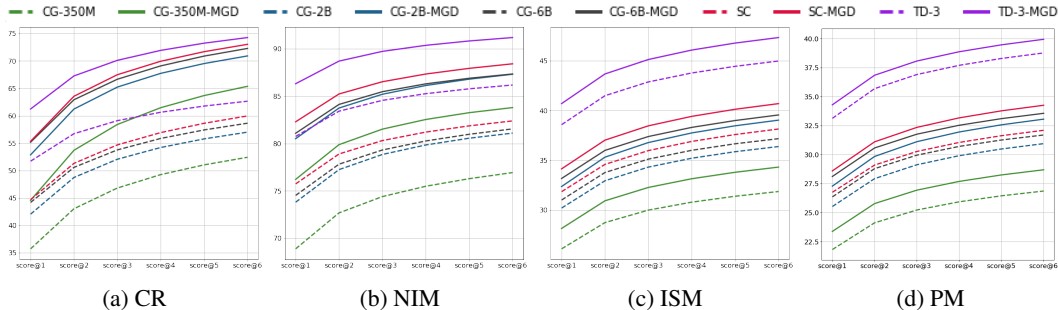

Figure 2: score@k for models with MGD and Standard prompt compared against base models. The values of $k \in [1, 6]$ are marked on the X-axis.

**Next Identifier Match.** As seen in Figure 2b, all the models with MGD outperform the respective base models, with a relative improvement of 7.11%-8.92%. The smallest model CodeGen-350M with MGD outperforms the much larger CodeGen-6B with a relative improvement of 2.76%. SantaCoder with MGD outperforms the larger CodeGen-6B by a relative margin of 8.42%, and even the largest model text-davinci-003 by 2.60%. TD-3-MGD outperforms TD-3 with a relative margin of 5.81%.

**Identifier Sequence Match.** Figure 2c shows that all the models with MGD outperform their respective base models on ISM, showing a relative improvement ranging between 6.41%-7.70%. SantaCoder and CodeGen-2B with MGD outperform the larger CodeGen-6B with a relative margin of 9.47% and 5.00% respectively. TD-3-MGD outperforms TD-3 with a relative margin of 5.24%

**Prefix Match.** Figure 2d shows percentage prefix match with ground truth. All the models with MGD outperform their respective base models with a relative improvement of 5.89%-6.82%. Both SantaCoder and CodeGen-2B with MGD outperform the larger CodeGen-6B with a relative margin of 8.08% and 4.30%. TD-3-MGD outperforms TD-3 with a relative margin of 3.03%.

**SC-MGD vs. TD-3.** SC is a 1.1B parameter model whereas TD-3 has 175B parameters. Being a much larger model and possibly due to other differences in training, TD-3 does better than SC across all metrics. Interestingly, with MGD, SC-MGD outperforms TD-3 on CR (Figure 2a) and NIM (Figure 2b). ISM and PM are method-level metrics and the relative advantage of the larger model prevails. Even then, with a small increase in the sampling budget, from $k = 1$ to $k = 4$, SC-MGD manages to surpass TD-3's performance with $k = 1$ on ISM (Figure 2c) and PM (Figure 2d).

**Summary.** MGD improves the compilability of code significantly, across architectures and parameter scale, leading even the smallest CodeGen-350M with MGD to outperform the largest LM, text-davinci-003. We also see improvements in all ground truth agreement metrics. Notably, smaller LMs with MGD outperform larger LMs (CodeGen-350M with MGD outperforms text-davinci-003 in CR and NIM, CodeGen-2B with MGD outperforms CodeGen-6B on ISM and PM) across all metrics.

## 4.2 Effect of MGD and Prompt Augmentation Strategies

We choose the best-performing base model, SantaCoder, from Section 4.1 to study the effect of prompting when combined with MGD. Figure 3 shows results for SantaCoder-{Standard, classExprTypes, RLPG} and TD-3, compared against SantaCoder with respective prompts and MGD.

**Compilation Rate.** Figure 3a shows the results for Compilation Rate. We observe improvements in compilation rate with both the prompting techniques, classExprTypes and RLPG, with RLPG marginally outperforming classExprTypes. We note that SantaCoder with Standard prompt and MGD is able to relatively improve over both RLPG and classExprTypes augmentation by 10.01% and 13.11% respectively. Further, SantaCoder with RLPG and MGD is able to outperform SantaCoder-RLPG and SantaCoder-classExprTypes with a relative margin of 17.70% and 21.02% respectively, while increasing the margin of relative improvement over text-davinci-003 to 24.70%.

**Next Identifier Match.** As seen in Figure 3b, similar to compilation, both RLPG and classExprTypes prompt augmentation leads to improvement over the base model. However, SantaCoder with either prompt augmentations underperforms text-davinci-003. SantaCoder with MGD outperforms

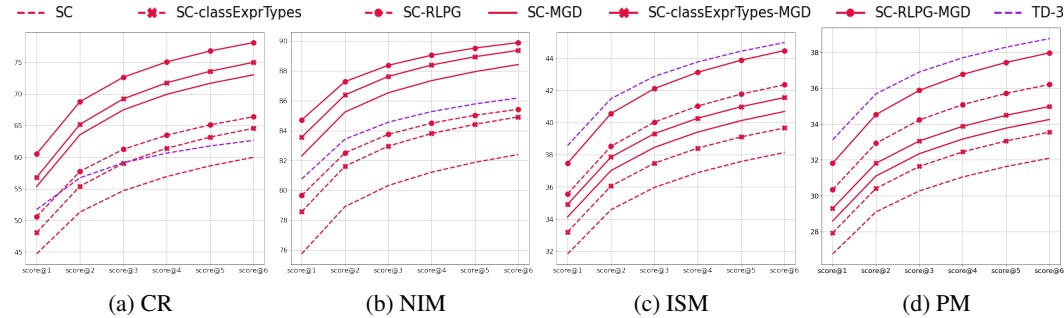

Figure 3: score@k for models with MGD and prompt augmentation compared against base models.

TD-3, and consequently, both SantaCoder-RLPG and SantaCoder-classExprTypes with a relative improvement of 2.60%, 3.51% and 4.14% respectively. SantaCoder with prompting and MGD outperform their respective baselines (SantaCoder with prompting) by a relative margin in the range of 5.23%-5.25%. We note that SantaCoder with RLPG and MGD increases the relative improvement with respect to the largest model, text-davinci-003 to 4.31%.

**Identifier Sequence Match.** On ISM, SantaCoder with prompt augmentation and MGD is able to outperform its respective baseline by a relative improvement of 4.78%-5.00%, while both the prompt augmentations result in an improvement over the base model. SantaCoder with RLPG and MGD is able to significantly reduce the gap with text-davinci-003, underperforming it by just 1.11%.

**Prefix Match.** As seen in Figure 3d, SantaCoder with prompt augmentation and MGD is able to outperform its respective baseline by 4.26%-4.87%.

**Summary.** While prompt augmentation techniques help in improving performance on all metrics, we see further improvement with MGD augmentation, and conclude that the contributions by prompt augmentation and MGD are complementary. Notably, SantaCoder-RLPG with MGD improves the relative margin for compilation rate with respect to text-davinci-003 to 24.70% compared to the 16.55% improvement achieved by SC-MGD, or 5.95% improvement achieved by SC-RLPG.

### 4.3 Effect of MGD on Fill-in-the-middle (FIM) Decoding

Among the base models, SantaCoder supports the FIM modality. We evaluated SantaCoder with autoregressive and FIM decoding strategies and text-davinci-003, and compared them with respective configurations of SantaCoder with MGD. Similar to our observations with prompt augmentation, while FIM modality leads to improvements across all metrics, we see continued improvement when using both FIM and MGD. Due to space limitations, detailed results are in Appendix E. Motivated by the complementary nature of FIM and MGD, we further evaluated SC-FIM-classExprTypes-MGD, combining both prompt augmentation and FIM modality. Consistent with our findings, it leads to a further improvement over SC-FIM-classExprTypes, as seen in Figure 5 (Appendix E).

### 4.4 Effect of Identifier Complexity on Next Identifier Match

Identifier names in repositories can get specific and long (Karampatsis et al., 2020). Due to this, while commonly used APIs may get tokenized into single tokens, identifiers specific to the context of individual repositories, especially in private settings, can span multiple subtokens in the LM vocabulary. We define the complexity of an identifier as the mean number of subtokens required to decode it. We show that the ability of LMs to accurately predict the identifier name decreases sharply with an increase in identifier complexity, while augmenting them with MGD improves their performance. MGD provides an improvement in the range of 21.79%-27.91% compared to the base model without MGD, across parameter scales, for the highest identifier complexity, which is prevalant in more than 36% of the methods in DOTPROMPTS. The detailed results are available in Appendix F.

# 5 Generalizability Study

We curate MGDMICROBENCH as a micro benchmark (10 examples), spanning 3 programming languages (Java, C#, Rust), 4 coding scenarios, and requiring use of 2 different static analyses, to evaluate generalizability of MGD. The detailed results are discussed in Appendix H.

**Different Programming Languages.** MGD utilizes static analyses that help infer and enforce semantic constraints on the code under development. Such analyses are available through the Language Server Protocol (LSP) (lsp) for most programming languages, e.g., clangd for C/C++ (cla, 2020), Jedi for Python (Roeca, 2019) and Rust Analyzer for Rust (rus, 2018). The monitor used in MGD can be instantiated as a thin client around LSP. Supporting new languages is easy and doesn't necessitate changes to the monitor's static analysis interface. Hence, MGD is applicable to most programming languages. Results on MGDMICROBENCH demonstrates MGD for Java, C# and Rust.

**Different Coding Scenarios.** A monitor under MGD triggers when a specified precondition is met during code generation. Flexibility in defining preconditions allows MGD to be applied to a variety of code scenarios, as follows: **1) Instantiation of valid classes:** A monitor triggers on 'new ', invokes a static analysis that identifies instantiable classes from the local & global context to ensure only valid classes are instantiated. **2)** `switch` **over** `enum`**:** A switch statement over enum uses named enums in `case <val>` to select branch in C# and Java. A monitor triggering on 'case ' is used to generate valid enums. **3) Correct number of arguments:** A stack-based monitor is implemented to guide for generation of right number of arguments to methods, that also handles nested function calls. **4) Joint monitoring for multiple properties:** Unlike previous examples that monitored for a single static property, we instantiate 2 joint-monitors: **a)** jointly guiding for type-correct dereferences & along with right number of arguments, **b)** jointly guiding for valid switch enum branches & type-correct dereferences. MGDMICROBENCH (appendix H.2) provides results over all of these scenarios.

**Different Static Analyses.** MGD is able to utilize results from various static analyses to guide the generation of code with LMs. We demonstrate MGD over the following properties in MGDMICROBENCH: **1) Typestates (Strom & Yemini, 1986):** Many APIs are stateful, and require callers to follow specific ordering of calls as part of the API contract. For example, a type representing a file handle would have a contract disallowing calls to read after close has been called. Such contracts can be expressed as finite state machines (FSMs), called typestates. **2) Session Types (Jespersen et al., 2015)** ensure that messages between concurrent programs are sent and received in the expected order, following a protocol, and are specified as communicating FSMs. In MGDMICROBENCH, we demonstrate a monitor for Rust that utilizes typestate and session-type design analyses.

# 6 Discussion

**Limitations.** Though static analysis is a mature field with strong theoretical foundations and several robust implementations of tools, analyzing partial and incomplete programs is still a difficult problem. In practice, editors such as Eclipse and Visual Studio support static analysis with heuristics. Though these heuristics are well-engineered and are widely used, they can be both imprecise (they can give incorrect suggestions) and incomplete (they can leave out correct suggestions). Satisfying functional-correctness specifications like pre/post-conditions and invariants is beyond the scope of this work. Consequently, though our results from guiding LMs using these analyses (through MGD) show improvements in quality metrics, additional steps such as testing and human inspection are needed to guarantee correctness of generated code.

**Societal Impact.** Software pervasively affects all aspects of our lives. With LMs being widely deployed as copilots and intelligent assistants to help developers write code, it is crucially important to develop tools like MGD to improve the quality of code generated by LMs (even if humans review and accept the suggestions given by LMs). Without such tools, we risk introducing bugs in code due to incorrect suggestions made by LMs, which has the potential to impact all of our lives negatively.

**Amount of Compute.** Our experiments do not involve any training, and we only perform inferences. We used machines of the following configurations: (1) CPU: 24-core AMD Epyc with 220GB RAM, GPU: Nvidia A100 80GB. (2) CPU: Intel Xeon(R) Platinum 8168 with 290GB RAM, GPU: Nvidia Tesla V100 16GB. For the experiments to evaluate text-davinci-003, we used the Azure API.

# 7 Related Work

**Pre-trained Models of Code.** Many powerful pre-trained models have been designed for code. These include encoder-only models like CodeBERT (Feng et al., 2020), GraphCodeBERT Guo et al. (2020) and CuBERT (Kanade et al., 2020); encoder-decoder models like PLBART (Ahmad et al., 2021), CodeT5 (Wang et al., 2021) and AlphaCode (Li et al., 2022); or decoder-only models like Codex (Chen et al., 2021), GPT-J (Wang & Komatsuzaki, 2021), Austin et al. (2021), GPT-Neo (Black et al., 2021), GPT-NeoX (Black et al., 2022), CodeParrot (Tunstall et al., 2022), PolyCoder (Xu et al., 2022), Incoder (Fried et al., 2022), CodeGen (Nijkamp et al., 2023), SantaCoder (Allal et al., 2023) and StarCoder (Li et al., 2023); or unified models like UniXCoder (Guo et al., 2022). Our monitor-guided decoding works only with logits and hence can be used with any model.

**Global Context of Code.** Hellendoorn & Devanbu (2017) build an n-gram model with a cache to track directory-level context. Xu et al. (2021) use locality based on directory-structure in retrieval-augmented modeling (Khandelwal et al., 2019). Many approaches use static analysis or previous generations (Zhang et al., 2023) to extract relevant code. They use the relevant context to either augment the prompt (Shrivastava et al., 2022; Pei et al., 2023; Zhang et al., 2023) or embeddings (Pashakhanloo et al., 2022; Ding et al., 2022) presented as input to the LM. Due to the limit on the prompt or embedding size, these approaches filter information through random walks (Pashakhanloo et al., 2022), classification (Shrivastava et al., 2022) or using fixed pruning strategies (Ding et al., 2022).

As we neither augment the prompt nor use extra embeddings, we do not need to prune the global context. We let the static analysis generate completion suggestions using the entire repository-level context. Zan et al. (2022) and Zhou et al. (2022) retrieve information from library documentation for prompt augmentation. Our static analysis analyzes libraries along with the repository-level source code. Several of these techniques require architecture modifications (Pashakhanloo et al., 2022; Ding et al., 2022) or finetuning (Zan et al., 2022; Ding et al., 2022; Pei et al., 2023). We use a simple interface between logits and static analysis with a frozen LM. Most of these approaches, excluding (Xu et al., 2021; Zhang et al., 2023), use one-time *a priori* retrieval. In contrast, we provide token-level guidance by invoking static analysis on demand. Our method is *complementary* to all the above approaches as they all try to condition the generation by modifying the *input* to the LM whereas we apply *output*-side constraints by reshaping the logits.

**Syntactic and Semantic Constraints.** There are two primary lines of work to enforce syntactic and semantic constraints on code generation, based on specialized modeling and through constrained decoding. GNN2NAG (Brockschmidt et al., 2019) and NSG (Mukherjee et al., 2021) are examples of the first and use attribute grammars. They are respectively evaluated on expressions that do not use user-defined methods or on methods with class-level context. We consider repository context for method-level completion. Unlike these approaches, our work is applicable to off-the-shelf LMs. PICARD (Scholak et al., 2021) and Synchromesh (Poesia et al., 2022) are constrained decoding approaches similar to ours. They use incremental parsing for syntactic validity and design domain-specific semantic checks to ensure semantic validity. Both are evaluated on SQL, and Synchromesh additionally considers domain-specific languages for visualization and calendar applications. In comparison, we target generation of general-purpose programming languages with focus on semantic constraints like type-consistency, API protocols, etc. using static analysis over repository context.

# 8 Conclusions and Future Work

In this work, we show how to use repository-wide information computed by static analysis (specifically, type-based analysis) using a stateful monitor as an interface, to improve quality of code generated by LMs. Our experimental results show the potential for significant quality improvements for code generation using this approach. Our approach is complementary to prompt augmentation techniques. It allows smaller models to achieve better or competitive performance compared to much larger models. This could open up the possibility of using smaller models directly within IDEs, alongside our monitor, as an alternative to the use of remotely-hosted Large LMs (LLMs), reducing inference costs and improving privacy. Our method is general and applicable to various coding scenarios where LMs are used generatively, such as code refactoring, code repair, or code completion, even if the repositories are in a transient state. We plan to expand the scope of MGD to more languages and deeper semantic analyses such as pre/post-conditions for which advanced constrained decoding methods (including backtracking and beam-search) might be needed.

## Acknowledgements

We thank the authors RLPG and of the LMs (CG, SC, TD-3) used in our work. We are grateful to Sheng Chen and Shi Chen from Microsoft, and the author of OLSP, Predrag Nikolic, for their generous help in answering our queries about interfacing with LSP. Our paper also benefited significantly from the constructive feedback by the reviewers.

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

# Appendix: Monitor-Guided Decoding of Code LMs with Static Analysis of Repository Context

**Outline**

## A    Monitor for the Running Example

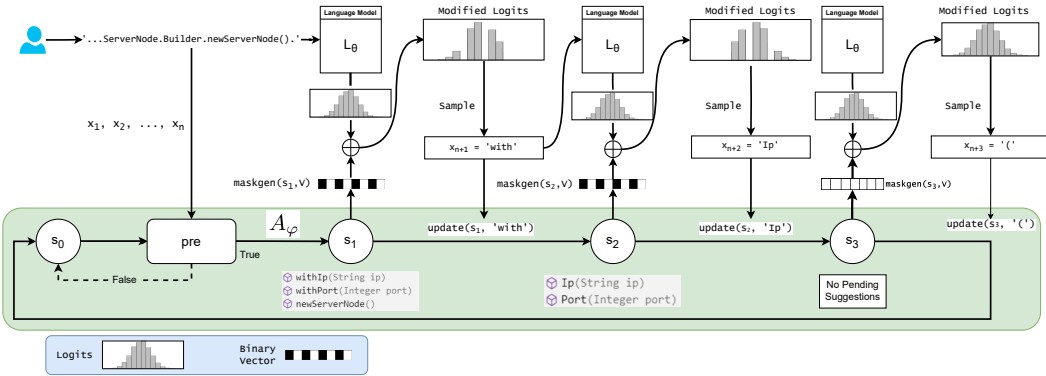

Figure 4: Monitor to guide generation of type-consistent identifiers for the code in Figure 1.

Figure 4 shows how the monitor interacts with the LM decoder for the example in Figure 1. Initially, the monitor, $M_\varphi$ is in the wait state $s_0$. Given the code completion input, $x_1, x_2, ..., x_n$, $M_\varphi$ first evaluates $pre(s_0, x_1, ..., x_n)$. Since $x_n =$ '.' (the dot symbol indicating object dereferencing in Java), $pre(s_0, x_1, ..., x_n)$ evaluates to $true$, and subsequently, in accordance with Eq. (4), the static analysis $A_\varphi$ is invoked, which determines the input prompt to be in accordance with the property $\varphi$, and resolves the type for the completion point to be `ServerNode.Builder`, as shown in the annotated AST in Figure 1(b). $A_\varphi$ then returns the set of identifiers consistent with the resolved type – {`withIp`, `withPort, newServerNode, ...`}, transitioning the monitor $M_\varphi$ to state $s_1$. $M_\varphi$ then calculates $m = maskgen(s_1, V)$, which masks out, for example, the token `host` (as inferred by SantaCoder in Figure 1(b)). Concurrently, the input is tokenized and the LM $L_\theta$ provides inferred logits, $\ell$ for the next token. The output logits $\ell$ from $L_\theta$ and mask $m$ are combined as $\ell \oplus m$ to obtain the modified logits, which is then softmaxed and a token is sampled –`with` in this case. The monitor then invokes $update(s_1, \text{with})$ to transition to $s_2$. Note that with the state transition, `newServerNode` is pruned from the set of identifiers, as the sampled token `with` does not prefix it. $L_\theta$ provides logits for the

next token, and the composition of $L_\theta$ and $M_\varphi$ is repeated to obtain modified logits, and finally we obtain the sampled token Ip. Since Ip is a prefix of a member in $s_2$, $update(s_2, \text{Ip})$ transitions to $s_3$, with $s_3 = \{\epsilon\}$, a singleton consisting of the empty string. Note that the token '(' is consistent with the suggestions in state $s_3$, since it matches the regular expression $w \cdot E \cdot \Sigma^*$, where $w = \epsilon$. Following the sampling of the token '(', the monitor transitions back to the wait state $s_0$.

## B  Data Set Curation Details

For the evaluation of pragmatic code generation, we require real-world repositories, with their complete environments and dependencies. Most public datasets for code generation do not meet this requirement, due to their focus on standalone code generation, except the recent CoderEval (Yu et al., 2023) and PyEnvs (Pei et al., 2023), both of which are not publicly available at the time of this writing. Further, CoderEval just evaluates 10 Java and 43 Python repositories. Since these datasets do not filter for repositories by their creation date, the test repositories could be a part of the training set for the evaluated LMs. Considering these, we describe the curation of PRAGMATICCODE, a dataset of real-world open-source Java projects complete with their development environments and dependencies. We ensure that the training repositories were released publicly only after the determined training dataset cutoff date (31 March 2022) for the CodeGen (Nijkamp et al., 2023), SantaCoder (Allal et al., 2023), and text-davinci-003 (GPT-3.5) (Ouyang et al., 2022) family of models. All the repositories in PRAGMATICCODE are buildable with their respective build systems. The development environment also provides support for analysis over templated code generated with systems like ProtoBuf (pro, 2008) and Lombok (lom, 2009). Further, we create DOTPROMPTS from PRAGMATICCODE for the evaluation of code generation in a pragmatic setting.

**PRAGMATICCODE.** We queried the GitHub API on 25 March 2023 to obtain a list of the top 1000-starred Java repositories on GitHub created after the determined cutoff date of 31 March 2022. We then attempt to download the latest snapshot of each of the top 1000-starred Java repositories and were able to download 731 repositories from GitHub. We create a build environment for Java projects consisting of Oracle Java Development Kit 17.0.6, Apache Ant 1.10.13, Apache Maven 3.8.7, Gradle 7.3.3, and GitHub CodeQL CLI 2.12.5. In this build environment, we invoke the CodeQL database create command[1] for every software repository obtained from GitHub. The CodeQL database creation process identifies the build system used in the repository and invokes the command for a clean build. The respective build systems use the project-level dependency information stored in configuration files like *pom.xml* and *build.gradle* to fetch the dependencies and store them locally. Next, we filter for repositories with permissive licenses, and filter out the repositories for which the CodeQL database creation failed, as that indicates that the repository either uses an unrecognized build system, some of its dependencies aren't satisfied, or the repository is in a transient state. We are left with 302 GitHub repositories after filtering for successful builds. We store the CodeQL database created for each of these repositories. Finally, we invoke the initialization of Eclipse JDT.LS[2] on each of the repositories, and filter for the repositories where JDT.LS could be successfully initialized. The final filtered list of 100 repositories, along with their CodeQL databases, comprise the PRAGMATICCODE dataset.

**DOTPROMPTS.** Yu et al. (2023) show that standalone functions account for less than 30% of open source projects among the top 100 most popular open source projects on GitHub, with most functions referencing third-party APIs or variables/constants defined in cross-file context. Hence, we create DOTPROMPTS for the evaluation of code generation in a pragmatic setting and aim to evaluate over real-world projects, not restricting to standalone function generation. Each task instance in DOTPROMPTS consists of a prompt up to a dereference location (dereference '.' operator in Java) within a target method, and the task is to complete the remainder of the method. Since dereference locations might be the points of occurrence for cross-file entities in source code, a model's ability to use cross-file context can be evaluated using DOTPROMPTS. **Curation Details.** We identify non-test class files that aren't auto-generated and have at least 1 cross-file dependency in the repository. From these files, we identify methods that aren't class and object initializers and have $\geq 2$ top-level statements spanning $\geq 7$ lines of source code, to ensure sufficient complexity in target methods. The average number of lines of code in the ground truth completion in DOTPROMPTS is 12.7. We use the CodeQL Query listed in section I to identify target methods from each of the repositories in PRAGMATICCODE based on the above criteria. As shown by Shrivastava et al. (2022), repositories are

---

[1]https://docs.github.com/en/code-security/codeql-cli/using-the-codeql-cli/creating-codeql-databases
[2]https://github.com/eclipse/eclipse.jdt.ls

quite uneven in terms of their size, so to avoid individual repositories from dominating our evaluation, we limit to including up to 20 methods from each repository as identified by the CodeQL query. In order to simulate the real-world usage scenario, where a developer may invoke code completion at different points within a method, we identify up to 10 uniformly distributed dereference locations within each of the identified methods. Each such dereference location becomes a data point for DOTPROMPTS.

## C  Experimental Setup - Additional Details

**Monitor Implementation.** Language Server Protocol (LSP) (lsp), is an open industry standard of communication between IDEs and programming language specific tools like static analyzers and compilers, called language servers. Eclipse JDT.LS (ecl, 2016) is a language server for Java, Rust Analyzer provides support for Rust, OmniSharp supports C# and JEDI supports Python. All language servers provide access to results of various static analyses over the same API. These can be accessed by implementing a Language Server Client. We implement an extensible and cross-platform language server client, `multilspy` with the aim to make it easy to setup and use these language servers in a language-agnostic manner. At the time of this writing, `multilspy` has been tested to work with language servers for Java, Python, Rust and C#, and we plan to extend the support to more programming languages and language servers.

Using `multilspy`, we specifically instantiate Eclipse JDT.LS as an engine that implements the static analysis $A_\varphi$ to check for type-consistency of identifiers. Notably, Eclipse JDT.LS supports reasoning with build time artifacts like those obtained from Project Lombok (lom, 2009) and ProtoBuf (pro, 2008), thus allowing it to consider a broad view of the code repository in the static analysis results it provides. We implement our monitor $M_\varphi$ as a thin layer around an LM, in accordance with Section 2, as a language server client that communicates with Eclipse JDT.LS. Since our implementation is based on LSP, and LSP is compatible with most major programming languages, our implementation can be easily ported to other languages besides Java, with the use of `multilspy`.

**Hyperparameters.** We use nucleus sampling (Holtzman et al., 2020) with a top-p value of 0.95 to generate 6 samples in total—1 each with temperature 0.2 and 0.4, and 2 each with temperature 0.6 and 0.8. We fix a prompt budget of (2048-512)=1536 tokens, and generation budget of 512 tokens, for a total context window size of 2048. If the text exceeds the prompt budget, we truncate it from the left for standard and classExprTypes prompts, and from the right for FIM. For classExprTypes augmentation with autoregressive decoding, we reserve a budget of 20% of total prompting budget for classExprTypes, and the remaining for autoregressive prompt. For FIM decoding without prompt augmentation, we reserve 50% of total prompting budget for the suffix. For FIM decoding with classExprTypes prompt augmentation, we reserve 20% of total prompting budget for classExprTypes, 40% for suffix, and the remaining for autoregressive prompt.

## D  Calculation of score@k

Let $S = \{s_1, s_2, ..., s_n\}$ be a multiset representing metric scores obtained across n-independent trials. Without loss of generality, we order $S$ in monotonic decreasing order as $S_\geq = (s_1^\geq, s_2^\geq, ..., s_n^\geq)$. We calculate $score@k, n$ as per the equation below:

$$score@k, n = \frac{1}{\binom{n}{k}} \sum_{i=1}^{n-k+1} \binom{n-i}{k-1} S_\geq[i] = \frac{1}{\binom{n}{k}} \sum_{T \in \binom{S}{k}} \max(T) \tag{5}$$

$$\binom{S}{k} = \{V | V \subseteq S, |V| = k\} \tag{6}$$

where $1 \leq k \leq n$, and $\binom{S}{k}$ is the set of all subsets of $S$ having cardinality $k$.

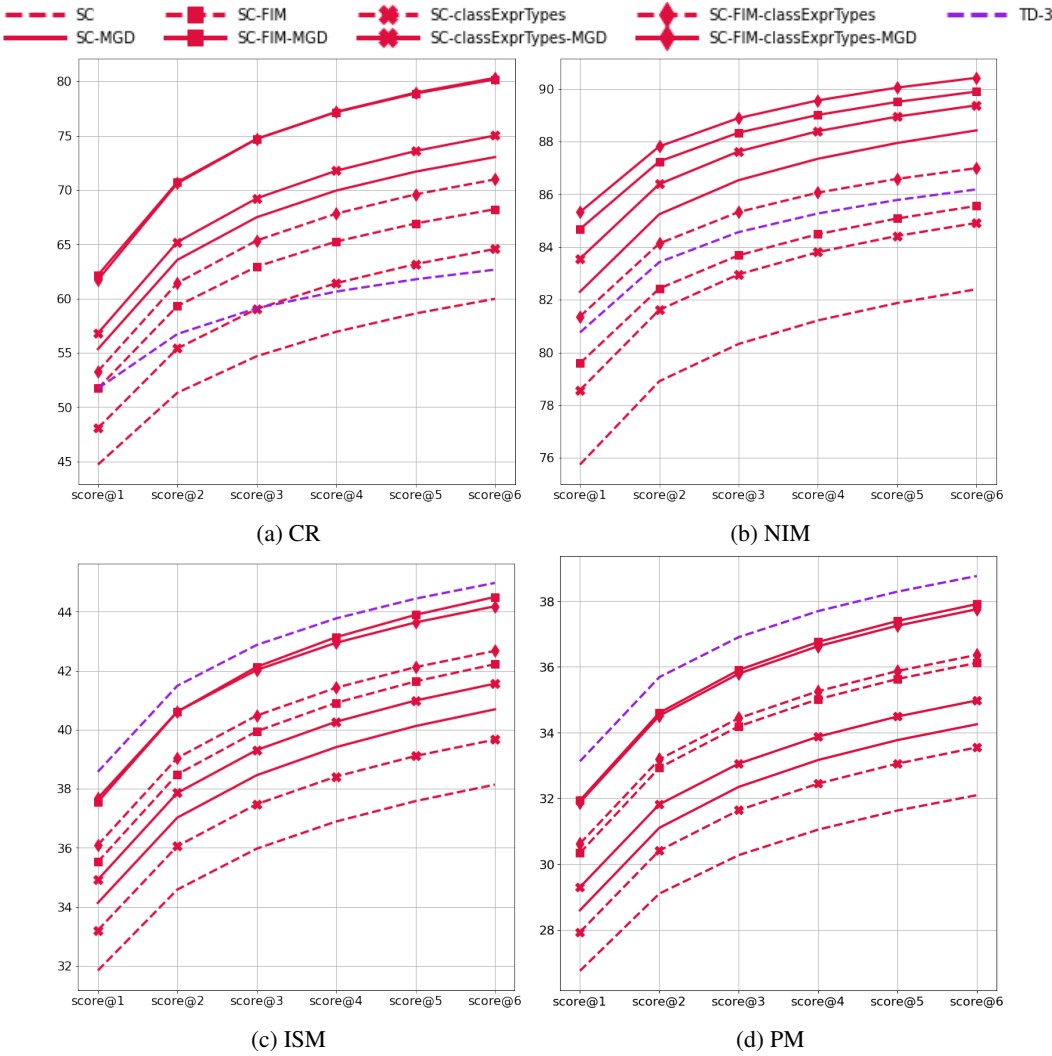

(a) CR

(b) NIM

(c) ISM

(d) PM

Figure 5: score@k for models with MGD and FIM compared against base models

# E  Effect of MGD on Fill-in-the-middle (FIM) Decoding - Complete Results

Among the base models, SantaCoder supports the FIM modality. Figure 5 shows the results for SantaCoder with autoregressive and FIM decoding strategies and text-davinci-003, compared with respective configurations of SantaCoder with MGD.

**Compilation Rate.** Figure 5a shows that SantaCoder with MGD outperforms SantaCoder-FIM by a relative margin of 7.04%. We see significant improvement in compilation rate, when SantaCoder-FIM is augmented with MGD, leading it to outperform text-davinci-003 with a relative margin of 27.97%. SantaCoder-FIM with MGD relatively improves over SantaCoder-FIM by 17.52%.

**Next Identifier Match.** Figure 5b shows that FIM modality boosts next identifier match but still underperforms text-davinci-003 and correspondingly, SantaCoder with MGD. Augmenting SantaCoder-FIM with MGD leads to a relative improvement of 5.07%.

**Identifier Sequence Match.** On ISM, SantaCoder-FIM with MGD improves over SantaCoder-FIM by 5.39%, which closes the gap with text-davinci-003, underperforming it by just 1.06%.

**Prefix Match.** Figure 5d shows that SantaCoder-FIM improves over SantaCoder, but still underperforms text-davinci-003 by a relative margin of 6.83%. SantaCoder-FIM with MGD outperforms

SantaCoder-FIM by 4.95% showing continued improvements, while reducing the gap with text-davinci-003, underperforming it by just 2.21%.

**Summary.** Similar to our observations with prompt augmentation, while FIM modality leads to improvements across all metrics, we see continued improvement when using both FIM and MGD.

**SC-FIM-classExprTypes-MGD.** Motivated by the complementary nature of MGD, we further evaluated SC-FIM-classExprTypes-MGD, combining both prompt augmentation and FIM modality, and consistent with our findings, it leads to a further improvement over SC-FIM-classExprTypes, as seen in Figure 5. We use classExprTypes with FIM since RLPG also selects a subset of post-lines in prompt augmentation in a large number of cases.

## F   Effect of Identifier Complexity on Next Identifier Match - Complete Results

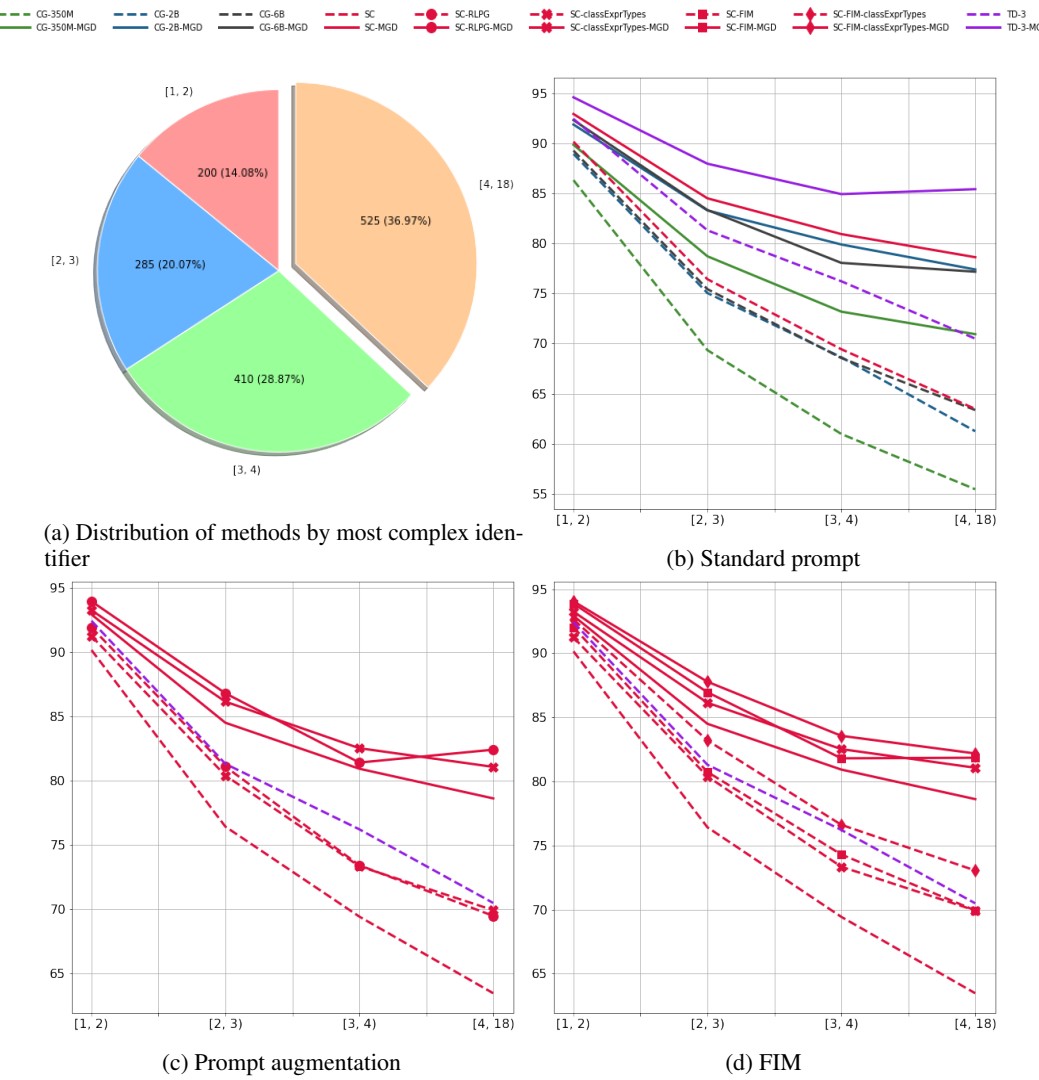

(a) Distribution of methods by most complex identifier

(b) Standard prompt

(c) Prompt augmentation

(d) FIM

Figure 6: (NIM, score@6) across next-identifier complexity

**Identifier Complexity.** Identifier names in code repositories can often get specific and long (Karampatsis et al., 2020). The vocabulary of LMs like CodeGen and SantaCoder is generally created by training a BPE tokenizer over the complete or a sample of the pretraining dataset (Sennrich et al., 2016). Due to this, while commonly used APIs may get tokenized into single tokens, identifiers specific to the context of individual repositories, especially in private settings, can span over multiple subtokens in the LM vocabulary, making their accurate generation difficult for the LM (both due to

Table 2: Statistics on inference time comparing CodeGen-6B and Codegen-6B with MGD. Time in seconds.

| | CodeGen-6B | CodeGen-6B-MGD |
|---|---|---|
| **Mean** | 22.57 | 41.34 |
| **Mean slowdown** | | 83.16% |
| **Standard deviation** | 12.44 | 22.39 |
| **Min** | 0.94 | 1.14 |
| **First quartile** | 21.23 | 26.76 |
| **Median** | 25.48 | 44.56 |
| **Second quartile** | 25.86 | 58.20 |
| **Max** | 79.06 | 111.73 |

the increased number of decoding steps leading to a larger search space and also due to the relative rarity of the identifier name). We define the complexity of an identifier as the mean number of subtokens required to decode it, using the tokenizers of all the models under study (CG, SC, TD-3). About 36.97% of methods in DOTPROMPTS dataset have at least 1 identifier of complexity [4, 18) as shown in Figure 6a, and therefore, for a model to generate those methods correctly, it is important for the model to be able to generate the complex identifier name.

**Next Identifier Match.** Figure 6 shows the result for the NIM metric, across increasing complexity of the next ground-truth identifier. We note that all models show a sharp degradation in performance with an increase in identifier complexity. The same trend holds true for prompt augmentation as well as FIM modality. Augmenting base models with MGD leads to significant improvement in the model's ability to predict the next identifier for the most complex identifier case [4, 18) with a relative improvement in the range of 21%-28%. The smallest model, CodeGen-350M with MGD achieves parity with the largest model, text-davinci-003 and outperforms the much larger CodeGen-6B with a relative margin of 11.95%. CodeGen-2B with MGD outperforms CodeGen-6B and text-davinci-003 by a relative margin of 22.14% and 9.79% respectively. SantaCoder with MGD improves over text-davinci-003 by a relative margin of 11.53%. We further observe that both prompt augmentation and FIM-modality with MGD lead to a diminishing rate of degradation, as can be seen in the curves for SC-FIM-MGD in Figure 6d. SantaCoder-FIM with MGD outperforms text-davinci-003 and SantaCoder-FIM by a relative margin of 16.11% and 17.04% respectively.

**Summary.** The ability of LMs to accurately predict the identifier name decreases sharply with an increase in identifier complexity. All models with MGD outperform their respective base models with a large relative improvement in the range of 21%-28%, with smaller LMs outperforming much larger LMs (CodeGen-350M-MGD achieves parity with text-davinci-003, and outperforms CodeGen-6B, CodeGen-2B-MGD outperforms CodeGen-6B). While prompt augmentation and FIM modality lead to improvement over baselines, they also suffer from a sharp decrease across identifier complexity, but augmenting them with MGD leads to similar large improvements as observed with base models (relative improvement in the range of 15.92%-18.59%).

# G   Impact of MGD on Inference Time

To study the impact of adding MGD on inference time, we compare the code generation times by CodeGen-6B model and CodeGen-6B with MGD. We select 500 prompts from DOTPROMPTS and generate up to 512 tokens with CodeGen-6B as well as CodeGen-6B with MGD. The inferencing is performed with HuggingFace implementation of the model on machine configuration (1) as described in Section 6. We use the same decoding scheme as described in Section 3. After generation, we filter out the cases where the number of tokens generated by CodeGen-6B and Codegen-6B with MGD is not the same, in order to ensure parity both in the size of the input prompt and number of generated tokens. We are left with 161 instances, where the size of the input prompt, as well as the number of generated tokens, is the same for both models. Figure 2 shows statistics related to inference time for both the models in seconds. We observe a mean slowdown of 83.16% in decoding time for CodeGen-6B with MGD. We note that there might be many opportunities to optimize our implementation though we have not explored them yet.

# H    Generalizability Study: Microbenchmark Results

Table 3 presents results over MGDMICROBENCH, detailing the scenarios we evaluated for studying the generalizability of MGD.

## H.1    Different Programming Languages

In MGDMICROBENCH, *CI1*, *NA1*, *VD1*, *J2* are coding scenarios in Java, *SE1*, *SE2* and *J1* are coding scenarios in C#, and *TS1*, *TS2* and *ST1* are coding scenarios in Rust for which monitors under the MGD framework, targeting the specific languages are built. Section 4 provided detailed evaluation of a monitor for Java. Hence, collectively, section 4 and MGDMICROBENCH demonstrate that MGD is viable for Java, C# and Rust.

## H.2    Different Coding Scenarios

**Instantiation of valid classes.** Scenario ***CI1*** (abbreviation for "class instantiation") considers demo code for a university setting, where `Person` is declared as an abstract class, extended concretely by `Student` and `Teacher` classes. In the scenario, the prompt code is setup to declare an object `p1` of type `Person` as follows: 'Person p1 = new '. The task for the LM is to generatively complete the code, assigning a concrete instantiation of `Person`. Following the keyword 'new', a constructor should be invoked. SC configuration, without MGD, invokes the non-existent constructor `Person(...)` (non-existent since it is an abstract class), whereas SC-MGD, with a monitor that differentiates between abstract and concrete classes invokes `Student(...)`, which is valid as per the scenario.

**switch over enum.** Scenarios ***SE1, SE2*** (abbreviation for "switch over enum") and J1 (abbreviation for "joint monitoring") are representative scenarios obtained from a real world emulation software written in C#. In scenarios *SE1* and *SE2*, the prompt code has been setup to 'switch(...)' over enum values of type 'AccessSize' and 'Intrinsic' respectively. The model configurations used are SC and SC-MGD. While SC generated the 'case' branch '1', which is a violation of the enum type, SC-MGD correctly generated the case branch 'AccessSize.Byte' which is type-correct while also being consistent with the ground truth. In *SE2*, while SC was able to get the type-name of the enum value correct, i.e., it generated 'Intrinsic', the enum value under the enum type it generated is non-existent, i.e., the symbol 'X86Comisdgt' does not exist, and hence invalid. SC-MGD generates a valid case branch, 'Intrinsic.X86Comisdlt', consistent with the ground truth. Scenario *SE2* is not completely solved due to the invalid dereference of '.LessThan' by SC-MGD, due to the absence of monitoring for dereferences, which is resolved in scenario *J1* below, in the discussion of joint monitoring.

**Valid number of arguments to methods.** Scenarios ***NA1*** (abbreviation for "number of arguments"), *VD1* (abbreviation for "valid dereferences") and *J2* are derived by modifying the underlying API presented in Figure 1a (in the main paper). The task is to use the `ServerNode.Builder` fluent API to create an instance of `ServerNode`. For these scenarios, we replaced individual methods `withIp(ip)` and `withPort(port)` (as seen in Figure 1a) with a single method `withIpPort(ip, port)` after which `build()` can be called to instantiate the `ServerNode` object. In scenario *NA1*, the prompt code has been setup with an open call to method `withIpPort(`, and the task for the LM is to generatively write code to pass the correct arguments to the method. SC configuration, without MGD generates a single argument, and closes the method call, thus violating the API of the `withIpPort` method. It further generates a call to the non-existent method `withPort`. SC-MGD configuration guided by a monitor for valid number of arguments, generates two arguments, corresponding to `ip` and `port` respectively, in-line with the contract of `withIpPort`. Further, it calls the method `build()`. In scenario ***VD1***, the setup is similar to *NA1*, except that the call to the method `withIpPort` is not made, and instead, the task for the LM is to generate the call to the right method `withIpPort`, and then generate the arguments for it. Unlike *NA1*, in this scenario, we use only the dereferences monitor. While the base configuration, SC without MGD, generates a call to non-existent 'hostAddress', SC-MGD with monitor for dereferences calls the right method, 'withIpPort'. However, without the monitor for right number of arguments, it just generates one argument, thus violating the API. It further calls `withIpPort` a second time, which is a type-correct dereference, however, not the expected response as per the ground truth, since it is a redundant call. Scenario *VD1* is not completely

Table 3: Results over MGDMICROBENCH

| ID | DESCRIPTION | GENERALIZATION ASPECT | WITHOUT MGD | WITH MGD |
|---|---|---|---|---|
| **Code scenario — Valid class instantiations** | | | | |
| *CI1* | University - Instantiation and assignment to abstract class | Code Scenario, Java | `Person p1 = new` `Person("John", ...)` | `Person p1 = new` `Student("John",` `...)` |
| **Code scenario — Valid enum constants within switch statements** | | | | |
| *SE1* | Emulation - switch over AccessSize enum | Code Scenario, C# | `case` `1` `: ...` | `case` `AccessSize.Byte` `: ...` |
| *SE2* | Emulation - switch over Intrinsic enum | Code Scenario, C# | `case Intrinsic.` `X86Comisdgt` `:` `... X86Condition.` `GreaterThan` | `case Intrinsic.` `X86Comisdlt` `:` `... X86Condition.` `LessThan` |
| **Code scenario — Correct number of arguments in method calls** | | | | |
| *NA1* | Fluent API - Parse Server | Code Scenario, Java | `arr[0])` `.withPort(...)` | `arr[0].trim()` `,` `Integer.parseInt(` `arr[1].trim())` `)` `.build()` |
| **Code scenario — type-correct dereferences** | | | | |
| *VD1* | Fluent API - Parse Server | Code Scenario, Java | `.` `hostAddress` `(arr[0])` `.port(Integer.` `parseInt(arr[1]))` | `.` `withIpPort` `(arr[0])` `.withIpPort(arr[1])` |
| **Joint monitoring for valid enum constants in switch and type-correct dereferences** | | | | |
| *J1* | Emulation - switch over Intrinsic enum | Code Scenario, C# | `case Intrinsic.` `X86Comisdgt` `:` `... X86Condition.` `GreaterThan` | `case Intrinsic.` `X86Comisdlt` `:` `... X86Condition.` `Below` |
| **Joint monitoring for type-correct dereferences and number of arguments** | | | | |
| *J2* | Fluent API - Parse Server | Code Scenario, Java | `.` `hostAddress` `(arr[0])` `.port(Integer.` `parseInt(arr[1]))` | `.withIpPort(` `arr[0].trim()` `,` `Integer.parseInt(` `arr[1].trim())` `)` `.build()` |
| **Static Analysis — Typestate API protocols** | | | | |
| *TS1* | Android MediaPlayer | Typestate, Rust | `stop();` | `reset();` |
| *TS2* | GPIO Pins | Typestate, Rust | `set_input_pull_up();` | `set_input_high_z();` |
| **Static Analysis — SessionType 2-party communication protocol** | | | | |
| *ST1* | Deposit money to ATM | SessionTypes, Rust | `recv();` | `send(0).recv();` |

solved due to the invalid number of arguments by SC-MGD, and will be resolved in scenario *J2* below.

**Joint monitoring for multiple properties.** MGD framework can utilize results from multiple monitors simultaneously to guide generation of code to follow multiple properties. Scenarios *J1* and *J2* explore 2 different instances of joint-monitors operating simultaneously. ***J1*** is the same coding scenario as *SE2*. In *SE2*, the monitor only monitored for generation of type-valid 'case' branches, whereas in *J1*, the monitor for valid case branches and monitor for type-valid dereferences are used jointly to perform the decoding. The difference can be noted in generation of the correct dereference 'X86Condition.Below' in *J1* (highlighted in green) for SC-MGD, which earlier was 'X86Condition.LessThan' in *SE2*, resulting in a non-existent symbol error. In *J1*, SC-MGD with the joint monitor is able to generate both the correct case branch, as well as a valid dereference. Scenario ***J2*** is the same coding scenario as *VD1*. In *VD1*, only monitor for valid dereferences was used in SC-MGD, whereas in *J2*, the monitors for valid dereferences and monitor for correct number of arguments to methods, are both used jointly with SC-MGD. The difference can be noted in the correct number of arguments (2) generated by SC-MGD in *J2*. Further, unlike in *VD1*, SC-MGD in *J2* subsequently calls build(), which is in-line with the ground truth, likely due to the improved code context that generating the right number of arguments provided.

### H.3 Richer Static Analyses and Constraints

Each of the different coding scenarios discussed in MGDMICROBENCH require a different static analysis to be performed, and hence, MGD is applicable with a variety of static analysis techniques. In this section, we focus on richer properties that can be enforced with MGD. Incorrect use of symbol names is a broad class of errors, and a root cause of many compile-time and run-time errors. Invalid use of defined symbol/method names can also lead to various compile-time and run-time errors, and hence, just having the symbols in context may not be helpful. For the following scenarios, we use the SC-FIM-classExprTypes and SC-FIM-classExprTypes-MGD configurations, which we found to be the strongest configurations due to the additional context, in the detailed evaluation in section 4. Consider the following scenarios:

**Typestate Protocols.** The Android MediaPlayer has complex constraints on the ordering of the API calls [3] which might be hard even for developers to reason about (Mishra et al., 2016). Runtime violation of such contracts lead to exceptions like 'IllegalStateException'. Mishra et al. (2016) proposes the use of typestate analysis (Strom & Yemini, 1986) to detect violations of such protocols, through static analysis techniques, and prevent the bugs at runtime. Duarte & Ravara (2021) show that typestate protocols can be enforced by the rich analysis supported in the Rust type system itself. We utilize this, and instantiate a monitor to guide LMs to generate typestate-valid API calls.

***TS1***: The task in scenario *TS1* (abbreviation for "typestate") is to generatively complete the partially-written code that iteratively play songs in a playlist using the Android MediaPlayer API. The completion is invoked at a point where the MediaPlayer object is in the Stopped state. While the base configuration generates a call to stop();, which is in violation to the API protocol, since the MediaPlayer is already in the Stopped state at the point of completion, the same model configuration with MGD generates reset(); which is the only legal transition at the state as per the implemented protocol.

***TS2***: Crichton (2023) and Rust on Embedded Devices Working Group et al. (2018) describe the representation of GPIO pin states and the transitions between them as a typestate protocol. We use the Rust API provided in Rust on Embedded Devices Working Group et al. (2018) and task the LM to complete partially written code, that initializes a GPIO pin and transitions to the input_high_z state. While the base configuration generates a call to set_input_pull_up(); which transitions to the input_pulled_high state, the same model augmented with MGD calls set_input_high_z(); which leads to the desired state.

**SessionType Protocols.** Session types can be used to capture ordering constraints on two party communication (Crichton et al., 2019) as communicating finite state machines. Jespersen et al. (2015) propose that Rust type system can be used to enforce session type contract at compile-time. We utilize the proposed approach, and build a monitor, that guides LMs to generate method-calls in line with the contract.

---

[3] https://developer.android.com/reference/android/media/MediaPlayer

***ST1***: Interaction with an ATM machine is an oft used example for two-way communication protocol. We use the session type formalism for ATM machine, described in (Jespersen et al., 2015). The task for the LM is to complete partially written code using the session type API, to deposit money and print the new balance. On the client side, after authentication and selecting the action to deposit money (from the ATM menu), the base model generated a call to `recv();`, which would wait for the ATM to send a value, whereas the protocol expects the client to "send" the deposit value. The base model augmented with MGD is able to correctly generate the call to `send(0).recv();`, which first sends amount to be deposited, and then waits for the ATM to communicate the new balance, which will be reported to the user. We note that with MGD, the model was able to generate code that follows the protocol appropriately.

# I   CodeQL Query for Identifying Evaluation Target Methods

```
/**
 * @id java/examples/find_target_methods
 * @name find_target_methods
 * @description Identify target methods from a Java repository along with
       classExprTypes information for DotPrompts dataset
 */

import java

predicate filterClass(Class c) {
        not(c instanceof TestClass) and
        c.fromSource() and
        not c.getFile().getRelativePath().matches("%generated%") and
        not(c.getFile().getRelativePath().matches("%test%")) and
        not(c.getFile().getRelativePath().matches("%target%")) and
        not(c.getFile().getRelativePath().matches("%build%")) and
        count(Method m1 | m1 = c.getAMethod() and filterMethod(m1) | m1) >= 1
}

predicate filterMethod(Method m) {
        m.fromSource() and
        not(m instanceof TestMethod) and
        not(m.hasName("<clinit>")) and
        not(m.hasName("<obinit>")) and
        m.getBody().getNumStmt() >= 2 and
        (m.getBody().getLocation().getEndLine() -
            m.getBody().getLocation().getStartLine()) >= 7
}

predicate typeDeclaredInFile(Type t, File file) {
        (t.fromSource() and t.getFile() = file) or
        (t.getErasure().fromSource() and t.getErasure().getFile() = file)
}

// Find classExprTypes files such that they contain type definition of any
    expressions defined in any callable in the class except for target_m
predicate filterClassExprTypeFile(Class c, Method target_m, File classFile, File
    classExprTypesFile){
        classExprTypesFile != classFile and
        (
                // classExprTypesFile contains type definition of a singly imported
                    type
                exists(Type t, ImportType impt |
                        impt.fromSource() and
                        c.getFile() = impt.getFile() and
                        t = impt.getImportedType() and
                        typeDeclaredInFile(t, classExprTypesFile)
                ) or
```

```
                    // classExprTypesFile contains type definition of return type or
                    //     param type of the target method
                    exists(Type t |
                            (t = target_m.getAParamType() or t =
                                target_m.getReturnType()) and
                            typeDeclaredInFile(t, classExprTypesFile)
                    ) or

                    // classExprTypesFile contains type definition of type of any
                    //     expression within a callable (that is not the target method) in
                    //     class c, or a return type of a callable or any of its parameters
                    exists(Expr e, Type t, Callable m |
                            m = c.getACallable() and
                            m != target_m and
                            (
                                    (
                                            e.getAnEnclosingStmt() = m.getBody().getAStmt()
                                                and
                                            e.getType() = t
                                    ) or
                                    t = m.getAParamType() or t = m.getReturnType()
                            ) and
                            typeDeclaredInFile(t, classExprTypesFile)
                    ) or

                    // classExprTypesFile contains type definition of any field type
                    exists(Type t, Field f |
                            c.getAField() = f and
                            f.getType() = t and
                            typeDeclaredInFile(t, classExprTypesFile)
                    )
            )
}

predicate expressionOfTypeContainedInBlock(Expr e, Type t, BlockStmt b) {
        e.getAnEnclosingStmt() = b.getAStmt() and
        e.getType() = t
}

from File classFile, File classExprTypesFile, Class c, Method m, BlockStmt b,
        int startLine, int startCol, int endLine, int endCol

where
        // Bind variables
        m = c.getAMethod() and
        b = m.getBody() and
        classFile = c.getFile() and

        // Apply filters
        filterClass(c) and
        filterMethod(m) and
        filterClassExprTypeFile(c, m, classFile, classExprTypesFile) and

        // Bind method boundary locations
        startLine = b.getLocation().getStartLine() and
        startCol = b.getLocation().getStartColumn() and
        endLine = b.getLocation().getEndLine() and
        endCol = b.getLocation().getEndColumn()

select
        classFile.getAbsolutePath(),
        classFile.getRelativePath(),
        classExprTypesFile.getAbsolutePath(),
        classExprTypesFile.getRelativePath(),
        startLine,
```

```
        startCol,
        endLine,
        endCol
```

## J   Examples of Code Generation with MGD

We present examples of code generation with MGD and compare them to generations without MGD augmentation. The appearance of red-markers below identifier names indicate that the identifier name is not type-consistent with the target object/class for the dereference operator. We highlight the correct identifiers generated with MGD augmentation in light green.

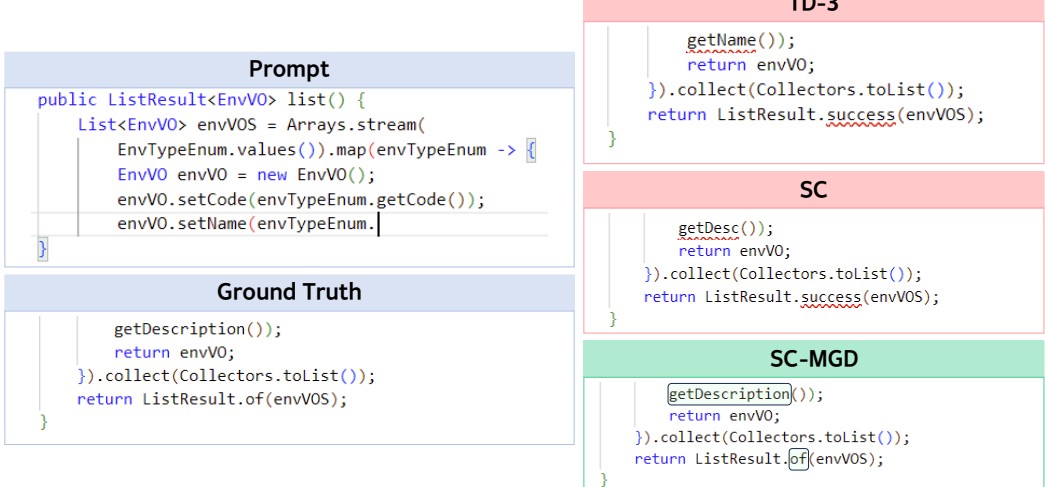

Figure 7: TD-3 and SC generate invalid identifier names: `getName`, `getDesc` for target object `envTypeEnum` and `success` for target class `ListResult`. Augmenting SC with MGD leads to the generation of correct identifiers: `getDescription`, `of` for the respective objects, leading to complete agreement with the ground truth.

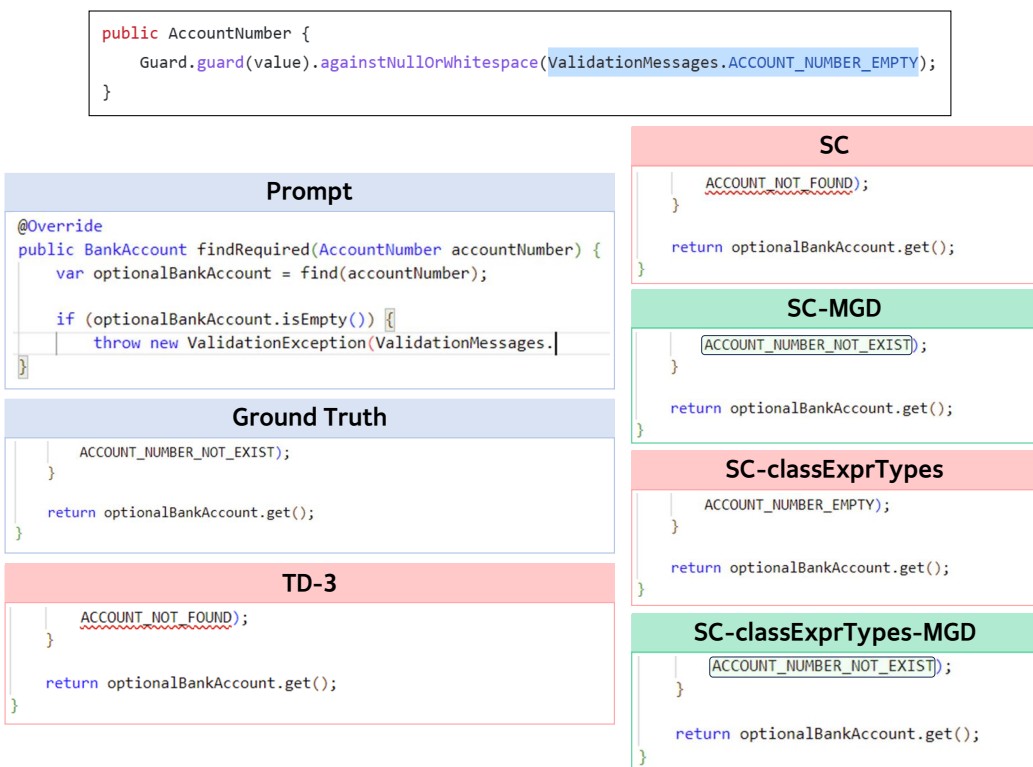

Figure 8: SC and TD-3 generate the identifier name: `ACCOUNT_NOT_FOUND`, which is invalid for the target class. The snippet of code presented above is from one of the files present in the classExprTypes augmentation. Augmenting SC with classExprTypes leads the model to generate the identifier name `ACCOUNT_NUMBER_EMPTY`, which is consistent with the type: `ValidationMessages`, and therefore compilable. However, it still does not match the ground truth. Augmenting both SC and SC-classExprTypes with MGD leads the model to generate the correct identifier name: `AC-COUNT_NUMBER_NOT_EXIST`, achieving agreement with the ground truth.

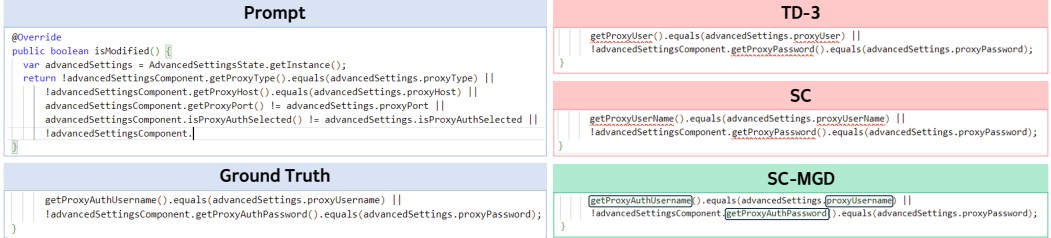

Figure 9: TD-3 and SC generate code that match the ground truth, except for the identifier names for target objects: `advancedSettings, advancedSettingsComponent`. Augmenting SC with MGD leads to generation of correct identifier names, and therefore agreement with ground truth.

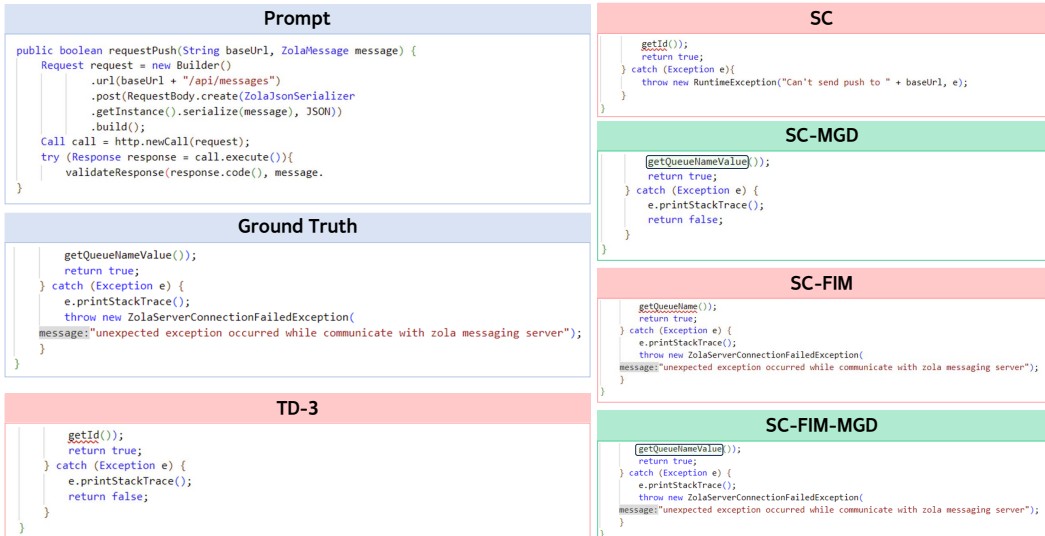

Figure 10: TD-3 and SC both generate code with invalid identifier names. SC augmented with MGD is able to use the correct identifier name and therefore generates compilable code. However, the generated code does not make use of the defined exception class: `ZolaServerConnection-FailedException` for exception handling. Augmenting SC with Fill-in-the-middle provides the model with the necessary context to perform exception handling, however, the model still hallucinates the identifier name `getQueueName` for the target object `message`. Hence, to get a correct generation, SC is augmented with both FIM and MGD, and this configuration is able to match the ground truth.

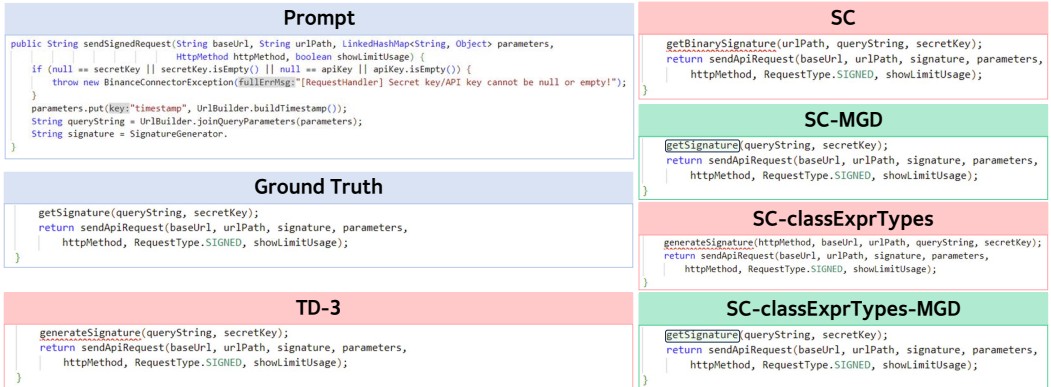

Figure 11: Example of generation with classExprTypes prompt augmentation.

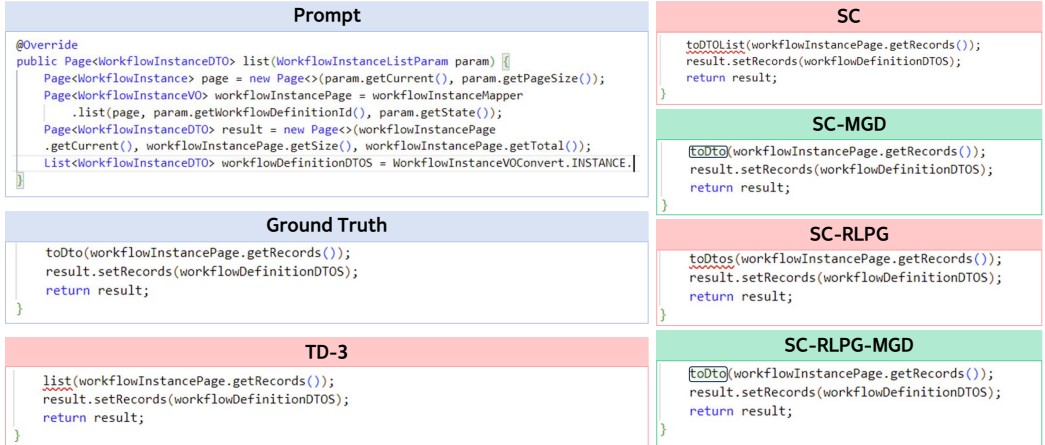

Figure 12: Example of generation with RLPG prompt augmentation. Augmenting SC with RLPG leads to generation of `toDtos` as compared to `toDTOList`, where both are invalid identifier names for the target object. Augmeting both the configurations with MGD leads to generation of the correct identifier and match with ground truth.

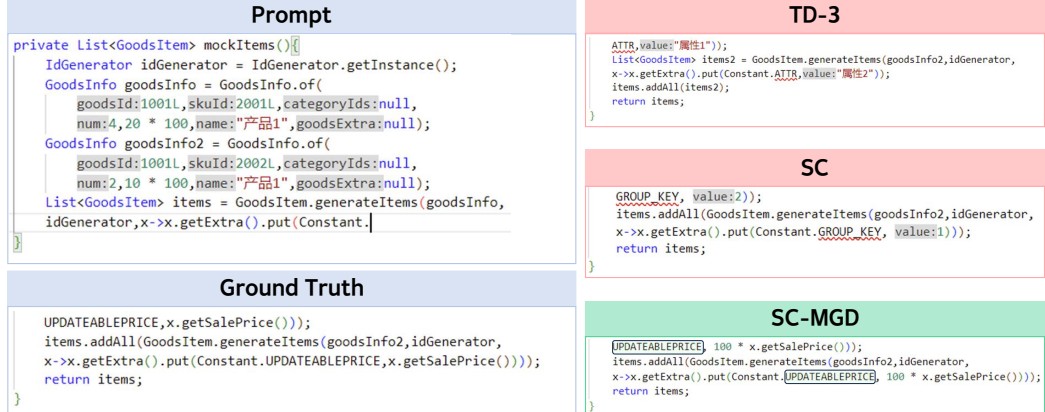

Figure 13: TD-3 and SC generate identifier names: `ATTR, GROUP_KEY` respectively for the target class `Constant`, which are both invalid. MGD augmentation leads to generation of correct identifier: `UPDATEABLEPRICE`.

