# OpenReview forum: "Monitor-Guided Decoding of Code LMs with Static Analysis of Repository Context"
_NeurIPS.cc/2023/Conference — NeurIPS 2023 poster_

### Official Review · Reviewer_27c5 · 2023-07-05

**Soundness:** 3 good
**Presentation:** 3 good
**Contribution:** 3 good
**Rating:** 7
**Confidence:** 4

**Summary:**

Authors propose to use static analysis to guide LLM decoding process to improve generation of code that may not be available in the context or training data. Authors show that their approach improves identifier generation compared to baseline and in certain cases compared to larger models.

**Strengths:**

- Authors propose a static analysis driven LLM decoding, which improves code generation (identifier generation) in situations when relevant code is not part of the context or training data
- Using static analysis to drive decoding rather than adding additional code information to the prompt/context, allows to preserve context for other needed information while improving the identifier generation
- Authors also create a PRAGMATICCODE dataset and DOTPROMPTS testset
- Monitor guided decoding shows significant improvement vs the same model without MGD.
- Authors also show the MGD is beneficial even when prompt augmentation is used.

**Weaknesses:**

- Unlike prompt/context augmentation, MGD approach can only improve output in specific cases where static analysis provides additional information. Although the paper shows significant improvement in the authors'  PRAGMATICCODE dataset and DOTPROMPTS testset, this testset is limited and geared towards identifier completion. Authors do not show a comparison in a general test set where identifier completion may be small part of accuracy. In such testset (and in real life usage), the MGD approach may not contribute much improvement. This would have to be known for real-life implementations.

I have read the author’s rebuttal.
Authors have addressed my concerns by explaining DOTPROMPTS testset and its metrics.

**Questions:**

- What is the cost/overhead of static analysis during decoding? How much does this slow down the result generation?
- In Table 1, could you explain the row ordering? Why CG is mixed with CG-X-MGD, but for SC, all SC-X-MGD lines are at the end of the table?

I have read the author’s rebuttal.
Authors have addressed my concerns by providing cost/overhead of static analysis and explaining the Table 1 row ordering.

**Limitations:**

It would be good to have a longer discussion of static analysis limitations. What static analyses can be applied and how costly they could be? As implemented MGD can only be applied for identifier completion. It would be good to know what other static analyses could be used and for what tasks.

I have read the author’s rebuttal.
Authors have addressed my concerns by further discussing static analysis techniques.

---

> ### Author Rebuttal · Authors · 2023-08-09
>
> > this testset is limited and geared towards identifier completion ... Authors do not show a comparison in a general test set where identifier completion may be small part of accuracy.
>
> We would like to clarify that each testcase in DotPrompts is obtained by identifying a dereference location in a method having at least 7 lines of source code, and the task for the model is to generate the rest of the method after the dereference (line 70-75 in Appendix). In DotPrompts, on average, the number of lines of code in the ground truth completion of a testcase is 12.7 and 75% quantile being 15 lines of code.
>
> Further, among the evaluation metrics, all metrics except NIM evaluate the complete method level generation by the model. These metrics are Compilation Rate (CR), Identifier Sequence Match (ISM) and Prefix Match (PM).
>
> As shown in our results, SOTA LMs struggle with the problem of generating type-consistent code, and our technique generates type-consistency without affecting (rather improving) match with the ground truth as measured by ISM and PM. Unlike contest style benchmarks, for example, HumanEval and CodeContests, we target pragmatic scenario encountered by real world developers in IDEs, where they are in the middle of implementing a method, in the context of a repository, rather than solving a standalone algorithmic problem.
>
> > What is the cost/overhead of static analysis during decoding? How much does this slow down the result generation?
>
> We acknowledge the importance of measuring the performance impact of MGD in real-time code generation. On average, the decoding time overhead of MGD compared to the same model without MGD is modest 1.83x (section G in Appendix). A tighter coupling with the Language Server Protocol architecture can further reduce this overhead.
>
> > It would be good to have a longer discussion of static analysis limitations ... As implemented MGD can only be applied for identifier completion. It would be good to know what other static analyses could be used and for what tasks.
>
> We posit MGD to be a general framework, which inherits the strengths and weaknesses of the static analysis techniques. MGD can be applied to many coding scenarios, like usage of correct number of arguments to methods and generic types, correct use of named-constant values, valid class instantiations, or to enforce richer code constraints like typestate or session type validity. All of these requires use of different static analysis techniques, but can be captured by MGD formalism. We have discussed these in the common response and will include them in the paper, including the experimental results on MGD-for-Rust case studies.
>
> However, to extend MGD to more sophisticated properties, such as general preconditions and post-conditions, advanced constrained decoding methods (including backtracking) might be needed. This is an exciting direction for future work and we will mention this in the main paper.
>
> > In Table 1, could you explain the row ordering? Why CG is mixed with CG-X-MGD, but for SC, all SC-X-MGD lines are at the end of the table?
>
> We thank the reviewer for highlighting this and realize that readability can be improved following your suggestion, and we will make the changes accordingly. At the time of writing, the rationale had been that each of the CG-X configurations pertained to different models (350M, 2B, etc.), whereas all of the SC-X configurations pertained to the same model, with augmentation in the space of FIM modality or prompting, and we resolved to keeping augmentation of the same model together, while separating different models. We will make the necessary changes.

---

> > ### Comment · Reviewer_27c5 · 2023-08-12
> > **Thanks to the authors**
> >
> > Thank you for addressing my questions and comments.

---

> > > ### Author Response · Authors · 2023-08-14
> > >
> > > We thank the reviewer for acknowledging that our response has addressed their questions and comments.

---

> > > > ### Comment · Reviewer_27c5 · 2023-08-17
> > > > **Raised rating**
> > > >
> > > > Based on all the author comments and explanations, I have raised my rating from 6 to 7.

---

> > > > > ### Author Response · Authors · 2023-08-17
> > > > >
> > > > > Thank you for revising the score!

---

### Official Review · Reviewer_fL4k · 2023-07-07

**Soundness:** 3 good
**Presentation:** 3 good
**Contribution:** 2 fair
**Rating:** 6
**Confidence:** 5

**Summary:**

The paper proposes a framework that modifies the output logits of a language model using a monitor-guided decoding approach. The monitor consists of type-guided method invocations across the repository obtained via static analysis tools and heuristics to update the contents of the monitor along with when to trigger it. The authors show performance improvements when their approach is used in conjugation with different LMs and when augmented with different prompt augmentation techniques.

**Strengths:**

- This paper is in line with some recent works that combine the strengths of static analysis tools (that can help incorporate domain knowledge and that are well-studied) with LLMs (that have inbuilt world knowledge).

- The paper is clear in writing and easy to understand.

- The experimental results are convincing.

**Weaknesses:**

- My main concern with the paper is the scope of its applicability to other settings. Even though the authors describe their framework as a general framework in Section 2, they have shown results in a very narrow setting of generating identifiers informed by type constraints. In the paper, the authors have used a pre-condition corresponding to the occurrence of the token "." which only represents method invocation in Java. Also, I do not see how the monitor will work in its current form when this pre-condition is made general, for example, say when the user hits a new line. It is not clear to me how this framework can be adapted to a more general code completion setting that is free of the notion of a pre-condition or to applications other than code completion (such as bug-repair) or even to a different programming language where it is difficult to define this pre-condition or the state corresponding to the monitor. I would like a discussion of the generalization capability of MCD to complex settings with concrete examples and experiments.

- I didn't see an analysis of the time and memory overhead due to MCD in the main paper. As per my understanding, this can be significant given that the static analysis tool is triggered at each invocation point and no form of caching is done to save time. I would like to know the time taken by MCD to generate an accurate prediction and see whether it is practical from the point of view of deploying it in conjugation with a real-time end-user application. This factor becomes increasingly important since the authors are performing repeated sampling to obtain gains.

**Questions:**

- Line 223-224: "For our experiments, we use n = 6 independent trials. For a budget of k ∈ [1, n] samples, we compute the aggregate score score@k..". Does this mean that you perform n forward passes through the LLM or sample n times? If the former, is the input to the LLM same during these passes?

- It is not clear to me as to what is the stopping criteria used for decoding. Can it generate multiple lines of code following the "." or is it a single line of code?

- Line 172-173: "The update function removes the members in s that are not prefixed by xn+1, and those prefixed by xn+1 are updated by pruning the prefix string xn+1." I didn't quite understand this.

- Line 206: " ..to adapt RLPG to DOCPROMPTS". Does this mean that RLPG was trained on DOCPROMPTS? Can you describe the process of adaptation and how the repository contexts selected by RLPG were incorporated with SantaCoder?

- Minor formatting issues: (a) The sentence is not clear at the end of the caption of Figure 1; (b) Extra spaces in between in the abstract.

**Limitations:**

Both limitations and potential negative societal impact are mentioned.

---

> ### Author Rebuttal · Authors · 2023-08-09
>
> > Main concern: scope of applicability to other settings. Authors describe framework as general ... have shown results in narrow setting of type constraints
>
> As discussed in the common response, MGD is generalizable across programming languages, coding scenarios as well as different static analyses. We emphasize that compilability is an important desirable characteristic of code, and we demonstrate that LMs across parameter scale struggle with generating compilable code. We show that generation of type-consistent dereferences plays a significant role in improving the compilability and match with ground truth of code generated by LMs.
>
> > "." as precondition ... only represents method invocation in Java. How monitor will work when pre-condition is made general. How framework can be adapted to general code completion setting free of notion of a pre-condition
>
> Our precondition check (pre) is syntax driven, which in general can be thought of as syntactic pattern matching. These can be made robust to whitespaces such as newlines. While we use `"."` as the precondition trigger for our monitor, other triggers are possible, for example, `"->"` and `"::"` in C++, `["new", " "]` or `["case", " "]` in Java, etc. These can be instantiated with a simple change to the precondition check (pre). It’s a language specific implementation which can be easily changed.
>
> > Application other than code completion (such as bug-repair) or a different PL ... Discuss generalization capability of MCD to complex settings
>
> MGD is agnostic to the downstream coding application, and can be used in all coding scenarios, where LMs are used generatively. For example, consider the development scenario of code refactoring, where the repositories are in a transient state. MGD can be useful to have the LM generate code using API names with the latest updates to the codebase. Further, we have several generalization aspects of MGD in the common response and will include experimental results on the same.
>
> > Time taken by MCD to generate ... is it practical for deploying in a real-time end-user application
>
> Yes, MGD is practical for real-time IDE deployment. Our implementation is based on Language Server Protocol, which are highly optimized implementations designed specifically for real-time IDE usage, and we inherit those optimizations. Specifically, we find that the end-to-end time overhead of generation with MGD is a modest 1.83x on average compared to a model without MGD (appendix G).
>
> > Memory overhead of MGD
>
> Language Servers used for MGD are already a part of IDEs like VSCode and Sublime Text, and loaded when a file of a particular language is opened. The instantiated monitor is a very thin client bridging the LM and the language server, and not expected to have any additional memory overhead beyond the language servers.
>
> > Since the authors are performing repeated sampling to obtain gains
>
> We clarify that we do not perform repeated sampling to obtain gains. MGD can enhance the quality of generation even with a single sample. Following the common practice of evaluating models with different sampling budgets, we allow each model to sample multiple generations and compare the models across different sampling budgets. Our results show that MGD helps across all budgets (in [1,6]) including a single sample for every model considered.
>
> Cost of sampling a single generation from larger models is higher than that of smaller models. Thus, it might be feasible to sample multiple generations from a smaller model at lower/comparable cost than generating a single generation from a larger model. It is in this context that we highlight that selecting from best of 3 samples of a 1.1B model (SantaCoder) with prompting and MGD, on average, has better prefix match with ground truth than much larger 175B model (text-davinci-003). This is in addition to showing that every model improves with MGD given the same sampling budget, including for one sample.
>
> Please let us know if further explanation is needed from our end about our setup.
>
> > Q1: Line 223-224 ...
>
> Given a testcase from DotPrompts, we use nucleus sampling with top-p value of 0.95 to generate $n=6$ samples using the same prompt (Line 93-95 in Appendix).
>
> > Q2: Stopping Criteria ...
>
> Yes, the LM is required to generate multiple lines of code following the dereference until end-of-method (On average, the number of lines of code to be completed in the ground truth completion of a testcase in DotPrompts is 12.7 and 75% quantile is as high as 15) with a budget of 512 tokens. The decoding terminates when a closing brace that matches the method’s opening brace is generated or the generation budget is exhausted.
>
> > Q3: Line 172-173 ...
>
> We explain this by an example. Kindly refer to figure 4 in Appendix which shows the decoding steps for the motivating example. Consider the transition from state $s_1$ = `{"withIp", "withPort", "newServernode"}` to state $s_2$ = `{"Ip", "Port"}`. Following one step of Monitor-Guided Decoding at state $s_1$, the token `with` is sampled. Next, `update` function removes all the strings in $s_1$ that can’t be generated following `with`: so `newServerNode` is removed. For the remaining strings, `update` truncates the part that has been sampled, so `with` is removed from `withIp` and `withPort`, leaving $s_2$ = `{"Ip", "Port"}`. We will improve the description in the paper.
>
> > Q4: RLPG ...
>
> RLPG considers a single line completion task (and hence a single line hole), whereas DotPrompts has a multi-line method-level generation task (and hence we consider the method suffix following an object dereference as a hole). We make necessary changes to adapt the retriever for the change in hole granularity. As RLPG model was trained for Java, and RLPG paper shows performance generalization to new repos, we reuse their officially released model checkpoints.
>
> We thank the reviewer for also suggesting formatting improvements and we will make the necessary changes.

---

> > ### Comment · Reviewer_fL4k · 2023-08-20
> >
> > Thank you for your thorough response! I am happy with the explanations provided by the authors regarding the generalization and applicability of MGD as well as the Rust experiments. I would say that setting the pre-condition for each language and thinking about what trigger tokens to use still requires some effort. For future work, I would encourage the authors to think of ways of removing this constraint. One way could be to let a classifier predict what trigger tokens to use for each language based on important keywords and the use-case.
> >
> > I would also recommend authors include a discussion about the stopping criteria, sampling budget, and memory/latency overhead in the main paper.
> >
> > RLPG: Can you please give details of how the RLPG retriever was adapted for multi-line completion?
> >
> > I have increased my score to 6 after reading the author's response.

---

> > > ### Author Response · Authors · 2023-08-21
> > >
> > > We are happy that our response addressed the reviewer's concerns. Thank you for revising the score and suggestions for future improvements. We will update the main paper as suggested by the reviewer.
> > >
> > > > RLPG: Can you please give details of how the RLPG retriever was adapted for multi-line completion?
> > >
> > > The RLPG implementation masks a single line whereas we mask all the lines following the object dereference (to prevent ground truth leakage into the RLPG-generated prompt). In our implementation, this is done through a simple wrapper around the released RLPG codebase. As RLPG model was trained for Java, and RLPG paper shows performance generalization to new repositories, we reuse their officially released model checkpoints. We will release our code. We note that our results show that MGD is complementary to prompt augmentation techniques like RLPG (section 4.2 in the paper).
> > >
> > > We would be happy to provide any additional details.

---

### Official Review · Reviewer_LEhe · 2023-07-07

**Soundness:** 3 good
**Presentation:** 2 fair
**Contribution:** 3 good
**Rating:** 6
**Confidence:** 4

**Summary:**

This paper proposes to use the output of a code static analysis tool, of the sort used in IDE code completion tools, to constrain the output of an LLM to improve code generation. Concretely, the paper focuses on type-consistency in object dereference for Java code. When an LLM is called to generate a use of an identifier from a class (i.e. using the '.' operator, in Java), the static analysis tool is invoked to only allow the LLM to generate identifiers accessible in that class which are consistent with the current typed context. This is implemented by masking the probabilities in the LLM to zero-out any tokens inconsistent with the set of valid identifiers returned by the analysis tool, and sampling from these constrained probabilities. The paper compiles a new dataset of open-source Java repositories with dependencies and development environments (to allow applying the static analysis tool), and a typed-dereference code completion task on this dataset. The method consistently improves the ability of black-box code LLMs (CodeGen and SantaCoder) to produce code that successfully compiles and matches the ground truth.

**Strengths:**

The approach is simple but well-motivated, and can be applied on top of black box code LLMs. I could easily see this approach being deployed effectively in IDEs, if it isn't already. As a type of constrained decoding technique, it should easily be compatible with approaches that e.g. augment the model prompt.

While the experimentation was limited to a single task and dataset, the evaluation on it was fairly thorough, evaluating two different base LLM families (CodeGen and SantaCoder) with FIM and non-FIM modes for SantaCoder, and comparing to reasonable baselines.

The approach demonstrates solid improvements across metrics, with large improvements in compilation rate and next identifier match.


**Weaknesses:**

The contribution felt a bit thin to me for a NeurIPS paper. I think that the general idea of using monitors is well-motivated and potentially widely applicable, but here they are only demonstrated on a single task which IMO is particularly well-suited task to the approach --- dereference constraints should really narrow down the space of compilable identifiers that the LLM can choose from, and the largest improvements from the method are on compilation rates and single identifier match. I think the paper would be much stronger if it showed that the approach can also be effectively applied on another task which affords static analysis, ideally one that LLMs are a clearer fit for (i.e. generating longer-form code, such as classes, while ensuring the generated code can compile), or one of the ones mentioned in the Future Work. This might require more sophisticated methods of integrating the static analysis constraints into the LLM decoding procedure beyond the simple prefix-consistency filtering used here.

I felt that the baselines could be improved. One missing baseline that I think is important would just choose randomly from the set of possible identifier candidates produced by the static analysis tool, without using an LLM at all. This seems like it could do pretty well on CR (although it's unclear to me how much code beyond the next identifier the task requires generating, see below), and potentially also on NIM (it would at least give a sense for how many identifiers are output by the tool). I was also pretty unclear on what the classExprTypes baseline is doing; see questions below.

The writing of the paper was somewhat unclear, and I had trouble resolving a number of important details, in particular about the experimental setup. The dataset and new task definitions could potentially be a contribution, but many details about them were unclear; see questions below.

--- Update after response ---
The author response largely addressed these concerns; see comment below.

**Questions:**

Q1) What static analysis tool is being used? Is it from some IDE (the discussion mentions Eclipse and Visual Studio), and if so which one?

Q2) What code does the model need to generate in the DotPrompts task, after the post-dereference location? Is it the rest of the method? Since this is a new dataset, it would help to provide some stats on how long the ground-truth completions are, to get a sense for their difficulty.

Q3) I didn't understand the classExprType approach from the explanation in line 202-205. What is being included in the prompt -- type definitions, or methods/expressions that have matching types, or something else?

Q4) It felt odd to me to not apply either MGD or the baseline methods to the text-davinci-003 model if I understood right, could some explanation be given for this (API costs)?

*Other comments* (not necessary to respond to in response)
- Clarify what sampling strategy is being used (e.g. top-p with a temperature? what hyperparameters?)
- I think the definition of pass@k in 225 is wrong -- it's the expected # of times that >=1 success occurs in the list of k candidates.
- Clarify how FIM interacts with truncation to fit within the context window.
- The description of the metrics in 211-221 was unclear to me, in particular how "ordered set" deals with repeated identifiers.
- The plots in Figure 3 were hard to read, with all lines being red.
- Figure 1 was a bit confusing --- do both text-davinci-003 and SantaCoder generate the same code (in red, with X)?
- "Basic concepts and notation" would be clearer if it's tied to the motivating examples, e.g. give a concrete example of property (compilation?)
- Lines 154-173 spend a lot of space describing the bookkeeping necessary due to vocabulary mismatches, but it's pretty wordy and I'm worried it might not make things clearer to someone who doesn't already have a sense of how this would need to be done. I think this could be moved to the appendix, or perhaps made more precise with an algorithm block (possibly in the appendix).


**Limitations:**

The paper should do a better job of indicating that the evaluation of the proposed monitor framework is pretty limited in the current work, evaluating only on a single task (and a single dataset). I'm worried that the token-level constrained decoding presented here might not work well when combined with more complex static analysis settings.

---

> ### Author Rebuttal · Authors · 2023-08-09
>
> > Q2) ... is it rest of the method?
>
> Yes, the task is to generate the rest of the method, starting from the dereference location. Since the methods in DotPrompts consist of at least 7 lines of source code, a typical completion consists of multiple lines (Appendix line 75). On average, the number of lines of code in the ground truth completion of a testcase is 12.7 and 75% quantile is as high as 15 lines of code.
>
> > Show applicability of MGD on task LLMs are a clearer fit for (generating longer-form code ... while ensuring generated code compiles)
>
> Among the evaluation metrics, except NIM, all other metrics - Compilation Rate (CR), Identifier Sequence Match (ISM) and Prefix Match (PM) evaluate the complete method-level generation. The significant jump in compilation in our results demonstrate that MGD guidance consistently improves ground truth match for longer form code generation, while significantly improving the compilability of the generated code.
>
> > experiments limited to single task & dataset, evaluation thorough ... different LLM families with FIM and non-FIM ... reasonable baselines.
>
> We thank the reviewer for recognizing the thoroughness of our evaluation. We would like to mention that DotPrompts was sourced from large number of real world repositories, with a long form code generation task. While metrics NIM and ISM capture the specific scenario of type-consistency, PM and CR are method-level metrics, used commonly in the literature, evaluating a broad range of real world coding scenarios and properties.
>
> > Paper would be stronger if showed approach can apply on another task that affords static analysis, or ones mentioned in Future Work
>
> Thank you for this suggestion. In the common response section, we demonstrate a concrete instantiation of MGD on a different programming language (Rust), utilizing 2 different static analyses (typestate & session-types), both of which are mentioned as future work. We will include these scenarios in the paper. We also show scenario for doing joint monitoring by combining multiple static analyses together (also mentioned in future work) for ensuring generation of type-correct method calls with right number of arguments passed (please refer to the common response).
>
> > How many identifiers are output by the static analysis?
>
> On average in DotPrompts, the static analysis returned 44.86 identifiers, with a median of 25 identifiers.
>
> > missing baseline: choose randomly from possible ... identifier candidates
>
> We implement the proposed baseline over TD-3 and evaluate it on a subset of DotPrompts with 533 test cases across 77 repos. Results are reported in **Table 1 and Figure 1 (TD-3-Random) in the attachment**. We observed a significant decrease across all metrics, likely due to the random sampling's inability to capture the intent implicit in the context, which the LLM captures.
> ## Questions
> > Q1)
>
> We use Eclipse JDT.LS, a Language Server for Java. It is an IDE agnostic tool and used to provide Java support in several IDEs (Appendix lines 84-92). As our implementation builds upon the Language Server Protocol, instantiating MGD for other languages can be achieved by changing the Language Server from Eclipse JDT.LS to another Language Server, as we did for the Rust PL discussed in the common response.
>
> > Q3)
>
> Given a testcase to complete method M, within class C - We mask out M from C, and run a static analysis in the remaining contents of C to identify all possible expressions in it (for example, dereference expressions, function calls, concatenation, etc.), and list the type of all such expressions. We then identify the files where each of these types are declared, and concatenate the content from those files, truncating to fit the context budget allocated.
>
> >  Q4)
>
> We thank the reviewer for the suggestion and implement MGD for TD-3. Detailed results are in **Table 1 in the attachment** and discussed in the common response.
> ### Other Questions
> > What sampling strategy is used ...
>
> We use nucleus sampling, with a top-p value of 0.95, generating 6 samples, 1 each at temperature 0.2 and 0.4, and 2 each at temperature 0.6 and 0.8 (Appendix line 93).
>
> > How FIM interacts with truncation to fit the context window
>
> We assign 20% of prompt token budget for ClassExprTypes, 50% for FIM, and truncate from left for standard and ClassExprTypes prompt, while truncating on the right for FIM prompt. In case of joint FIM and ClassExprTypes, we assign a budget of 20% for ClassExprTypes, and 40% for FIM (Appendix lines 95-102).
>
> > ... how "ordered set" deals with repeated identifiers
>
> We thank the reviewer for suggesting an improvement in writing of the evaluation metrics. We are using “ordered sequences” and not “ordered set”. We will update it in the writeup. Since we are using sequences, repeated identifier names are also matched for. For example, if ground truth identifier sequence is ["withIp", "withPort", "withIp", "withIp"], and the model generates the sequence ["withIp", "withPort", "withIp"] or ["withIp", "withPort", "withIp", "withPort"], then it gets a score of 3/4, but if it generates ["withIp", "withPort", "withIp", "withIp"], then it gets a score of 4/4.
>
> > Fig 1 ... text-davinci-003 and SantaCoder generate the same code?
>
> Yes, both text-davinci-003 and SantaCoder generate the same code present in the red box with X.
>
> > "Basic concepts and notation" ...
>
> > Definition of pass@k ...
>
> > The dataset and new task definitions could potentially be a contribution ...
>
> We thank the reviewer for their suggestions and for highlighting the dataset as a contribution. We will incorporate your valuable feedback. We will:
> 1. Adapt the discussion in section A of Appendix, to be clearer, and fit with the "Basic concepts and notation" section of the paper.
> 2. Rectify the wording on pass@k metric.
> 3. Rectify the dataset section to explain details related to the dataset, including relevant statistics like typical length of testcases as reported here and Appendix section B.

---

> > ### Comment · Reviewer_LEhe · 2023-08-13
> >
> > Thanks to the authors for the extremely thorough response, which addressed my major concerns! The Rust experiments are a definite strength that help show the generality of the approach. I'm also much more convinced that the approach can be applied to a broader range of tasks given the clear examples of trigger words and the LSP in the general response. It was helpful too to clarify that the Java task does involve method-level completion, and to show that the method outperforms the random identifier baseline and gives improvements on TD-3.
> >
> > I still have a bit of a concern that the constrained decoding method used might not work well when trying to integrate MGD into tasks where the static analyzer constrains the tokens a substantial distance into the generated output (if I understand right, the Java tasks constrain at the beginning of the generated output, and the Rust setting seems to involve short outputs), but this could likely be addressed through a backtracking or beam-search-like approach.
> >
> > I've raised my score to a 6 (from a 4).

---

> > > ### Author Response · Authors · 2023-08-14
> > >
> > > We are glad that our response addressed the reviewer's major concerns and thank the reviewer for revising their score accordingly.
> > >
> > > We agree with the reviewer that advanced decoding schemes using backtracking, beam-search or look-ahead could be combined with MGD, particularly to target richer constraints. It is an exciting future direction to pursue. Thank you!

---

### Official Review · Reviewer_hw3j · 2023-07-07

**Soundness:** 2 fair
**Presentation:** 3 good
**Contribution:** 2 fair
**Rating:** 5
**Confidence:** 4

**Summary:**

Inspired by the fact that IDE static analysis helps in code writing, this work proposes an MGD approach that uses a monitor to guide the decoding generation of the code language model. The authors started with the motivation that the code language model generation process does not perceive the global information of the repository thus leading to hallucination errors. The proposed MGD method uses static analysis to obtain the global context of the repository scope instead of the local context and uses this monitor as a stateful interface between LM. Experimental results show that the MGD method can significantly improve the quality of Java code generation.


**Strengths:**

1. Compared to introducing global information through prompt engineering, language model architecture modification, incremental training, etc., MGD is a simpler and more effective way of limiting the output by utilizing construct masks. However, such an idea is not innovative.
2. The experimental results given are good and can support some of their claims, especially the effectiveness of the MGD method in type-consistent identifiers.

**Weaknesses:**

1. Constructing masks to constrain the output space using grammar information, compilation information, and code definition information is classic in the code generation, Text-to-SQL Task. The authors also mention in related work that Text-to-SQL work such as PICARD has a similar constraint strategy, but MGD is targeted at general-purpose programming languages and focuses on static analysis through repository-level context.
However, MGD cannot be theoretically justified to support general-purpose programming languages, or even to effectively limit output in only some coding scenarios. In this work, the MGD utilizes the "." as a trigger to update the state, is it possible to find a single unique trigger for all coding scenarios, are there scenarios to consider consecutive multiple words as triggers, or spanning words as triggers? Also, can MGD support nested programming languages? Again, this would require setting up special triggers, and complex state management.

2. The experimental design, including the construction of the dataset and evaluation metrics, appears to be specific to
monitoring type-consistent identifiers. I understand that this is an instantiation of the MGD method, but my concern is that the MGD method only supports special instantiations from the methodological idea and experimental validation.

**Questions:**

As mentioned in the weaknesses, can MGD be extended to other programming languages and other coding scenarios?

**Limitations:**

The authors discuss the limitations of their work and give ideas for improvement.

---

> ### Author Rebuttal · Authors · 2023-08-09
>
> > Authors mention ... MGD is targeted at a general-purpose programming languages and focuses on static analysis through repository-level context. However, MGD cannot be theoretically justified to support general purpose programming languages, or even to effectively limit output in only some coding scenarios
>
> We have demonstrated that MGD does support general purpose programming languages (as shown by large improvements in both compilation and method level match for Java, in a dataset consisting of real world projects), however, we target specific properties that can be checked statically and enforced during decoding, like those supported by type systems. We have provided examples of additional coding scenarios and static analyses (typestate and session-types) in the common response. We have also built MGD for the Rust programming language and evaluated it on a mini-benchmark, whose results (in the attached PDF) show that MGD can guide LMs in generating code that adheres to richer properties.
>
> We acknowledge that functional correctness specified through pre/post conditions and invariants may not be enforceable through MGD in the current form. We will make these limitations explicit in the paper, and thank the reviewer for pointing them out.
>
> > Experimental design, dataset construction and evaluation metrics appear to be specific to type-consistent identifiers. I understand that this is an instantiation of the MGD method, but my concern is that the MGD method only supports special instantiations from the methodological idea and experimental validation.
>
> We clarify that the task in DotPrompts is to generate the complete method following a dereference location, with an average ground truth completion in the testset being about 12.7 lines of code. Hence, the task is long form code generation.
>
> Compilability is a necessary characteristic of any code and a general evaluation metric. In this paper, we demonstrate that LMs across parameter scale struggle with generating compilable code. While Next-Identifier Match (NIM) and Identifier Sequence Match (ISM) capture the specific scenario of type-consistency for MGD, Prefix Match (PM) and Compilation Rate (CR) are method-level metrics, that are used commonly in the literature, and are not specific to MGD.
>
> We show that generation of type-consistent dereferences plays a significant role in improving the compilability of code generated by LMs (relative improvement  of 22.67% over SantaCoder and 22.81% over text-davinci-003 with MGD). While MGD is used in this paper to target a specific problem (generating type-valid dereferences), it is shown to improve match with ground truth (PM) as well. In the common response, we have discussed generality of MGD along several dimensions.
>
> > In this work, the MGD utilizes the "." as a trigger to update the state, is it possible to find a single unique trigger for all coding scenarios
>
> As correctly pointed out by the reviewer, for different coding scenarios, the triggers may be different; however, the same MGD framework can support all of these triggers. For example, for object instantiation in Java, the trigger could be `["new", " "]`, and for named `enum` values in switch statements, the trigger could be `["case", " "]` (details in the common response). In both cases, the function pre in the formalism can be modified to handle these triggers.
>
> > Can MGD support nested programming languages? ... require setting up special triggers, and complex state management.
>
> We believe the reviewer is referring to nested functions or nested classes (kindly correct us if our interpretation is wrong). Languages with advanced type systems, that support these features can be leveraged by MGD. For example, Java and C# have such advanced features, and their Language Servers are able to provide type consistent identifiers even with nesting. Further, our datasets PragmaticCode and DotPrompts indeed contain such evaluation scenarios. The monitor instantiated in the paper already handles this scenario, including nested function calls and definitions. We will highlight these interesting cases with concrete examples in the Appendix.

---

> > ### Comment · Reviewer_hw3j · 2023-08-21
> >
> > Thanks for the response from the authors. I would like to clarify that I agree that the MGD method is effective in its proposed programming languages and scenarios. What I would like to discuss with the authors is whether the MGD method can be reused for languages with other programming paradigms. I try to give two examples of SQL code generation here.
> >
> > > SELECT AVG(salary)
> > FROM employees
> > WHERE salary > (
> >     SELECT budget / 2
> >     FROM departments
> >     WHERE name = 'Engineering'
> > );
> >
> > This one simple example seems to involve a number of questions. The nested query of the above SQL statement is supposed to return a scalar value, can the MGD method handle that?  The MGD method does not seem to have nested state management, how does it return to the previous level after ending the nesting? "(" can be used as a trigger to limit the fields followed by the AVG aggregator and also to trigger a nested query, is there a conflict?
> >
> > > SELECT T2.name, COUNT(*) FROM cocert AS T1 JOIN stadium AS T2....
> >
> > This is also a simple example. T2 is an alias, if MGD utilizes "." as a trigger can it guide the subsequent generation to stadium columns, currently "stadium" is not generated.
> >
> > Also, can the authors explain more about the innovativeness of the MGD method compared to past methods that used external structural information to guide code generation?

---

> > > ### Author Response · Authors · 2023-08-21
> > >
> > > We thank the reviewer for providing us the opportunity to discuss generalizability of MGD to other programming languages, coding scenarios and usage of richer static analyses. Following their review, we have discussed different coding scenarios (for example, class instantiation), extension to static analyses (for example, typestate and session-types), and programming languages (for example, Rust) in our response. Thank you for acknowledging the effectiveness of MGD in the proposed programming languages and scenarios.
> > >
> > > > The nested query of the above SQL statement is supposed to return a scalar value, can the MGD method handle that?
> > >
> > > As discussed in the paper and rebuttal, MGD inherits the strengths and weaknesses of the underlying static analyses. In case of SQL, several static analyses are available through the Language Server Protocol (with which our monitor interfaces). For this specific example given by the reviewer, we created a schema for `employees` and `departments` tables based on the example given by the reviewer. We then check whether given the table name, a static analysis could suggest schema-valid identifier names. Specifically, given the prompt `SELECT AVG(salary) FROM employees WHERE salary > ( SELECT departments.`, a static analysis for SQL returns `[budget:INT, name:VARCHAR]`, which can then be used to select only the type-valid identifiers. This shows that nested expressions in SQL can be handled by MGD using the static analysis.
> > > For the next scenario, prompting the language server with `SELECT T2.name, COUNT(*) FROM departments AS T1 JOIN employees AS T2 on T2.` returns `[name:VARCHAR, salary:INT]`, and hence, MGD will be able to handle aliasing in SQL variable names as well.
> > >
> > > > how does it return to the previous level after ending the nesting?
> > >
> > > This is a great question. Handling arbitrary nesting will require a stack-based monitor. We are implementing one to handle the case of monitoring for correct number of arguments to methods (where each argument itself can be a nested subexpression) - a scenario described in the common response above. Please note that with appropriate definitions of the `pre` and `update` functions, handling nesting is possible within the MGD formalism introduced in the paper.
> > >
> > > > can the authors explain more about the innovativeness of the MGD method compared to past methods that used external structural information to guide code generation?
> > >
> > > We broadly fall in the space of constrained decoding techniques, however, the use of rich static analysis (beyond structural properties, for example, typestate) for constraining code generation on-the-fly is novel. As discussed in the “Related work” section, structural information has been used for generating code in domain specific languages like SQL, SMCalFlow and Vega-Lite. We target general-purpose programming languages (like Java and Rust), and bring the benefits of years of static analysis research and LSP based IDE integration to code generation using LMs. Other works that also try to constrain general purpose programming languages include GNN2NAG and NSG. Unlike GNN2NAG and NSG, we do not need specialized architectures/modifications or additional training to guide the models with results of rich static analyses. This immediately makes our technique applicable to off-the-shelf LMs which we think is the main and pragmatic innovativeness of our method. In the paper, through extensive experimentation we have shown this to be true for multiple models (CG, SC, TD-3) of varying parameter scales (350M-175B). Further, MGD nicely complements prompt augmentation techniques such as RLPG, which have been specifically designed to capture repository-level context.

---

### Author Rebuttal · Authors · 2023-08-09

We appreciate the reviewers' constructive feedback and suggestions. We first answer some common questions and present individual responses later.

# Generalization and Applicability of MGD
We demonstrate MGD’s applicability to more coding scenarios, programming languages, and different static analyses & properties. We will include this discussion in the paper.
## Generalization Dimensions
### Programming languages
MGD can be applied to most programming languages (PL). For instantiation, it requires static analyses that help infer and enforce semantic constraints on the code under development. Such analyses are available in IDEs (e.g., clangd for C/C++, Jedi for Python & Rust Analyzer for Rust) through the standard Language Server Protocol (LSP). We build a monitor as a thin client around LSP (Appendix lines 84-92).

Supporting new PLs is easy and doesn’t necessitate changes to the monitor's static analysis interface, as it is a generic LSP client. *We were able to develop an MGD implementation for Rust using Rust Analyzer in just two hours following reviewer comments* (discussed below).

### Coding scenarios
MGD can be extended to several coding scenarios, including those requiring spanning token sequences as triggers. Example scenarios realizable with MGD formalism introduced in the paper are as follows:
1. **Class instantiation**: MGD can trigger on a 2-token span: `['new', ' ']` to ensure only valid classes are instantiated. The trigger invokes a static analysis that identifies instantiable classes from the local & global context at the trigger location.
2. **Method call and arguments**: As an example of joint monitoring based on multiple static analyses, consider a monitor for 2 properties, $M_a$) Type-consistent dereferences (from the paper) and $M_b$) Correct number of arguments to calls. $M_b$ triggers on the final state of $M_a$ when the last token contains `'('` (start of a call). A static analysis determines numbers of arguments that decoded method takes and $M_b$ states correspond to the number of arguments left to be decoded. `update` function transitions on arguments to the current function call, accounting for nested parentheses. $M_b$ prevents the generation of a token with `')'` (end of the call) till the right number of arguments have been decoded.
3. **`switch` over `enum`**: A switch case statement over `enum` uses named enum values in `case <val>` to match. A monitor with `pre` triggering on multi-token sequence `['case', ' ']` is used to generate valid named values.

### Static analysis techniques
We now provide examples of deeper semantic properties (& static analyses) that can be used with MGD (concretely instantiated for Rust as discussed next):
1. Typestates [1], often expressed as finite state machines (FSMs), define valid sequences of operations that can be invoked on objects of a given type. For example, a type representing a file handle would have typestate that disallows calling `read` after `close` has been called.
2. Session Types [2] ensure that messages between concurrent programs are sent and received in the expected order, following a specified protocol, and specified as communicating FSMs.

## Concrete instantiation of MGD for Rust with Typestate & Session Type consistency
To concretely show that MGD can be applied to other programming languages & use results from other static analyses, we instantiated MGD for Rust using Rust Analyzer [3]. A mini-benchmark with small code completion scenarios requiring conformance to typestate or session-type specifications for valid generation along with results is reported in **Table 2 of the attachment**. Despite using the strongest SantaCoder configuration from our paper, SC-FIM-classExprTypes, the generated code doesn’t satisfy the typestate and session-type properties (e.g., generating `stop()` which is not valid in the state `Stopped` of `MusicPlayer`). Use of MGD ensures generation of correct invocations. We will include these results in the paper.

# Additional Results
Following reviewer LEhe’s suggestion, we instantiate MGD over text-davinci-003 (TD-3), and a valid identifier random selection baseline. Detailed results are in **Table 1 & Figure 1 of the attachment**. Notably, TD-3-MGD achieves a compilation rate of **73.73% (22.81% improvement over TD-3)** and all other metrics see significant improvements as well. TD-3-Random suffers across all metrics, likely due to random sampling's inability to capture the intent implicit in the context, which the LLM captures.

# Performance Overhead
Language Servers are optimized for real-time use in IDEs. MGD integrates with Language Servers (which perform static analysis) and inherits the optimizations such as caching and reuse of pre-computed values. In our implementation, MGD's average decoding time overhead is a modest 1.83x (appendix G). In production setting, tight integration with Language Servers can reduce the overhead further.

# Generality of Dataset and Metrics
DotPrompts testset task requires generating the *complete method after the dereference location* spanning multiple lines. Test cases are derived from methods with $\geq7$ lines of code (Appendix, lines 70-75). Mean ground truth code length to be generated is 12.7 lines, with a 75% quantile of 15 lines, making DotPrompts a method-level code generation task. The LM is allowed to decode upto 512 tokens and MGD monitors every dereferenced identifier during decoding.

All evaluation metrics except NIM assess method-level generation: Compilation Rate, Identifier Sequence Match, and Prefix Match. To achieve a successful compilation, the model not only has to generate a type correct next identifier, but all subsequent method calls and field accesses must be valid as well.

[1] Strom et al., Typestate: A programming language concept for enhancing software reliability. IEEE TSE 1986

[2] Jespersen et al., Session Types for Rust. In WGP 2015 (pp. 13-22). ACM

[3] “Rust Analyzer.” (online)

---

### Decision · Program_Chairs · 2023-09-21

**Decision:**

Accept (poster)

**Comment:**

In this paper, the authors aim to address the problem of code LLMs in a larger code context that contains use types or functionalities defined in another module or library. The authors introduced the concept of monitors, which use static analysis in the background to guide the decoding. The authors tested this approach in generating identifiers that can improve compilation rates and match with the ground truth.

The reviewers appreciated the motivation and novelty of the approach and its effective way to limit output generation simply through masking. While this idea is not technically challenging, the overall approach is quite a simple and elegant way to constrain the output search space of code LLMs, showing clear performance gains in experiment results. There were concerns from reviewers about the generalization of this approach to other types of tasks rather than generating type-consistent identifiers. However, the authors' rebuttal has mostly addressed these concerns, and I highly encourage the authors to include more discussion in their final paper.